# Essential and dual effects of Notch activity on a natural transdifferentiation event

Thomas Daniele [1,2], Jeanne Cury [1], Marie-Charlotte Morin [1], Arnaud Ahier [1,3], Davide Isaia [1,4] & Sophie Jarriault [1] ✉

Cell identity can be reprogrammed, naturally or experimentally, albeit with low frequency. Why some cells, but not their neighbours, undergo a cell identity conversion remains unclear. We find that Notch signalling plays a key role to promote natural transdifferentiation in *C. elegans* hermaphrodites. Endogenous Notch signalling endows a cell with the competence to transdifferentiate by promoting plasticity factors expression (*hlh-16/Olig* and *sem-4/Sall*). Strikingly, ectopic Notch can trigger additional transdifferentiation in vivo. However, Notch signalling can both promote and block transdifferentiation depending on its activation timing. Notch only promotes transdifferentiation during an early precise window of opportunity and signal duration must be tightly controlled in time. Our findings emphasise the importance of temporality and dynamics of the underlying molecular events preceding the initiation of natural cell reprogramming. Finally, our results support a model where both an extrinsic signal and the intrinsic cellular context combine to empower a cell with the competence to transdifferentiate.

During metazoan development, cells become increasingly specialised as they acquire a final identity that fulfils a specific function in a particular tissue or organ. Nevertheless differentiated cells have been shown to naturally change their identity in many different species and tissues[1,2], although demonstration at the single-cell level is often lacking. The process whereby a cell stably and fully switches from one differentiated identity to another has been called transdifferentiation (Td) or direct reprogramming[3,4]. Examples include the conversion of pigmented cells into lens cells during lens regeneration in different species[5], the conversion of kidney distal tubule epithelial cells into an endocrine gland in zebrafish[6], or, in mammals, the conversion of heart venous cells into coronary artery cells[7].

Direct reprogramming can be experimentally reproduced by forcing the expression of specific transcription factors in differentiated cells in vivo or in vitro[8–10]. Experimental models of induced Td typically exhibit a fairly low success rate. Thus, within a population of similar cells, only a few will actually switch identity. Most induced cells will not undergo Td, making it difficult to focus on the mechanisms

involved, let alone predict which cells will transdifferentiate. Furthermore, a number of studies have shown that for a given inducing cue, certain cells types, including fibroblasts, neuroblastoma or liver cells, can be reprogrammed but not others[11–13]. These pioneer studies on natural and induced direct reprogramming have highlighted important questions that remain unanswered. Why is only a small subset of cells within a population able to transdifferentiate? How is a cell endowed with the capability of changing its identity, and particularly for natural Td events, to what extent do extrinsic cues and context, such as the micro-environment, or the intrinsic cellular content, modulate this capability? Here we have taken advantage of a single-celled Td event, which naturally occurs in 100% of animals and can be followed in vivo, to address these issues. We report evidence that an extrinsic signal through the Notch pathway is involved in potentiating in vivo Td by promoting intrinsic reprogramming factor expression.

The Notch signalling pathway is a highly conserved pathway involved in a plethora of essential developmental processes including cellular plasticity. It plays a role in cell-fate determination and

[1]Department of Development and Stem Cells, Institut de Génétique et de Biologie Moléculaire et Cellulaire (IGBMC), CNRS UMR 7104, INSERM U1298, Université de Strasbourg, Illkirch, France. [2]Present address: Vertex Pharmaceuticals (CH) GmbH, Zug, Switzerland. [3]Present address: Queensland Brain Institute, The University of Queensland, Brisbane, QLD, Australia. [4]Present address: Skyhawk Therapeutics, Basel, Switzerland. ✉e-mail: sophie@igbmc.fr

differentiation or tissue homoeostasis by regulating apoptosis or stem cell maintenance[14,15] and mediates both lateral or inductive signalling[16]. Notch signalling studies have been pioneered in invertebrate model systems such as *Drosophila melanogaster* and the nematode worm *Caenorhabditis elegans* where the Notch receptor was first identified[17–20]. Mechanistic aspects of Notch signal transduction, which involves the interaction between a receptor-expressing cell and a ligand-expressing cell, are well understood[21]. The contact between the Notch receptor(s) and its ligand(s) induces a succession of proteolytic cleavages of the receptor that lead to the release of the Notch intracellular domain (NICD). The NICD translocates to the nucleus and then interacts with the DNA-binding protein CSL (CBF1/RBPjκ/Su(H)/Lag-1) to activate the transcription of target genes[21–27].

Notch signalling has been shown to be necessary and sufficient to specify a rectal cell named "Y", in *C. elegans*[17]. In absence of Notch signal, no Y cell is made[17,28]. The Y cell is one of six fully differentiated rectal cells forming the rectal tube, and the only one with an extraordinary behaviour: it subsequently migrates away from the rectum during mid-larval development and switches its identity to become a motoneuron, named PDA[28,29] (Fig. 1A). This process has been the focus of in-depth studies at the single cell level and unambiguously shown to be a stereotyped bona fide Td event[28,30–33]. Intriguingly, the other rectal neighbouring cells, seemingly identical, never change their identity: what makes the Y cell special?

Here we show that Notch signalling not only specifies the Y cell as rectal but also endows this cell with the ability to change its identity. Strikingly, experimentally-provided, ectopic, Notch activity is sufficient to create, in vivo, another cell with the potential to transdifferentiate. RNAi and expression analyses showed that the extra Td utilises the same molecular mechanisms as in the natural Y-to-PDA Td[31,32,34] and highlighted a set of genes downstream of the Notch signalling that are strictly necessary for Y cell reprogramming and not for Y rectal identity. By supplying prolonged or transient Notch signalling to the Y cell at different time points, we demonstrated that seeing the Notch activity at the wrong time can block Y Td by re-enforcing its rectal identity. Thus, while a Notch signalling is crucial to allow a cell to change its identity, out-of-time Notch can block Td. Finally, the careful analysis of the characteristics of the supernumerary PDA neurons and the cells they originate from, showed that (i) this ectopic Td requires a favourable cellular environment to occur; and (ii) that Notch signalling levels do not impact when Y Td occurs. Together, our findings highlight that Notch signalling dynamics and tight regulation in vivo are essential to endow a unique cell with the capacity to transdifferentiate.

## Results

### Notch activity is sufficient to endow a cell with the capacity to transdifferentiate

A second Y cell is found in the specific Notch gain-of-function (*gf*) allele, *lin-12^Notch^(n137)*[17,28]. We had previously shown that this second Y cell transdifferentiates into a second PDA neuron later in development[28] (Fig. 1A). To understand the role of Notch signalling in Y-to-PDA Td, we first investigated if the generation of more PDA neurons is observed in other *lin-12^Notch^(gf)* alleles. We found that all *lin-12^Notch^(gf)* alleles examined [i.e. *n950* or *n302*, strong and milder *lin-12^Notch^(gf)* alleles respectively] exhibited an extra PDA neuron (Fig. 1B). Concomitantly, a neuron named DA9 is also absent every time that a second PDA is found (Fig. 1C). In wild type (WT), after its final division, the ABplpppaaaa blastomere [hereafter termed DA9^prog^, for DA9 *progenitor* to signify the early undifferentiated state of the cell that will become the future DA9 neuron] differentiates into the DA9 neuron. In *lin-12^Notch^(gf)* mutants, DA9^prog^ is mis-specified into a supernumerary Y cell[17,28] that later transdifferentiates into a supernumerary PDA neuron[28] (Fig. 1C, D). Together with the fact that extra Y cells made in other mutant backgrounds (e.g. *mab-9* or *egl-38* loss-of-function mutants) do not have the capacity to transdifferentiate into a PDA

neuron[28], our results suggest that Notch signalling not only promotes a rectal Y fate, but also confers the competence to transdifferentiate. By contrast, *lf* or *gf* alleles of *glp-1*, the second *C. elegans* Notch receptor gene, had no impact on the number of PDA neuron, indicating that GLP-1^Notch^ is likely not involved in Y Td (Supplementary Fig. 1A).

We next examined if Notch activity is sufficient to obtain supernumerary PDA neurons. To this aim, we over-expressed ectopically a Notch IntraCellular Domain (NICD) protein, a constitutively active form of the LIN-12^Notch^ receptor, in vivo. First, NICD was expressed under the *lin-12^Notch^* promoter, provoking the activation of the Notch pathway only in the cells that normally express *lin-12^Notch^*. Strikingly, worms carrying the *lin-12^Notch^p::NICD* transgene had 2 PDA neurons, instead of 1 in WT, and an associated absence of the DA9 neuron, a phenotype similar to *lin-12^Notch^(gf)* alleles (Fig. 1E). Next, we investigated whether additional Tds (i.e., more than one PDA) could also be observed when we exposed most cells in the worm to activated Notch. To this aim, a heat shock (hs) promoter, driving broad expression of NICD across tissues, was used[35]. Remarkably, we found that a ubiquitous pulse of ectopic Notch signalling at the time of Y birth also results in the formation of an extra PDA, indicative of an extra Td (Fig. 1F). Therefore, an embryonic pulse of Notch signalling is sufficient to later produce an additional Td in live animals. Altogether, these results indicate that Notch signalling is sufficient to create cells competent to transdifferentiate, in vivo.

To further investigate when and where Notch signalling acts to promote an additional Td, we determined the expression of *lin-12^Notch^* in the rectal area: a *lin-12^Notch^* transcriptional reporter[36] is expressed in the ABprpppaaaa cell [hereafter called Y^prog^ to describe the progenitor state of that cell, after its final division and before it has differentiated into the rectal Y cell], and in the DA9^prog^ cell, from their birth to approximately the twofold stage in WT embryos (Fig. 2A, Supplementary Fig. 1B). After this point, no residual signal was detected in Y^prog^ or DA9^prog^, showing that *lin-12^Notch^* transcription takes place only during a restricted time window. Analysis of a *lin-12^Notch^* translational reporter, known to rescue *lin-12^Notch^ lf* mutants[37], showed that the presence of the LIN-12^Notch^ receptor follows the exact same dynamic pattern as the *lin-12^Notch^* transcriptional reporter in the Y^prog^ cell (Fig. 2B, Supplementary Fig. 1B, summarised in Fig. 2C), in accordance with previous reports that the LIN-12^Notch^ protein has a very short half-life[38]. Together with the phenotype of the *lin-12^Notch^(gf)* mutants, these results suggest that LIN-12^Notch^ signalling could act within the Y^prog^ or DA9^prog^ cells to endow them with the competence to transdifferentiate.

### *hlh-16* is required cell-autonomously for the initiation of Y-to-PDA Td

We next examined what factors mediate this additional Td. During our initial genetic screen to identify the molecular players involved in Y-to-PDA Td, we retrieved the *fp12* allele[30], which results in a high penetrance of "0 PDA" defects. *fp12* had been mapped using EMS-density mapping[30], which we confirmed by SNP mapping and found to affect the *hlh-16/Olig* gene (Supplementary Fig. 1C), a nuclear bHLH protein[39]. *fp12* bears a point mutation (G → A) in the exon 2 splicing acceptor site, leading to a *lf* mutation. In *hlh-16(fp12)* mutant, a very high penetrance of "0 PDA" defects is observed (Fig. 3A). Two experiments confirmed that *fp12* is a *hlh-16 lf* mutant: (i) RNAi experiments against *hlh-16* phenocopied *hlh-16(fp12)* mutant (Supplementary Fig. 1D) and (ii) overexpression of WT *hlh-16* under a *hlh-16* promoter in the *hlh-16(fp12)* mutant background rescued the "0 PDA" phenotype (Supplementary Fig. 1E). Further analyses revealed that, in *hlh-16(fp12)* mutant, the Y cell is born and differentiates into a rectal cell as in the WT (Supplementary Fig. 1F). However, it then remains rectal at its original position, with its characteristic morphology and expressing the *egl-5* rectal marker. Td is never initiated in the L2 stage (Fig. 3B, C). These results suggest that *hlh-16/Olig* is implicated in Y Td initiation, rather than in Y cell-fate determination.

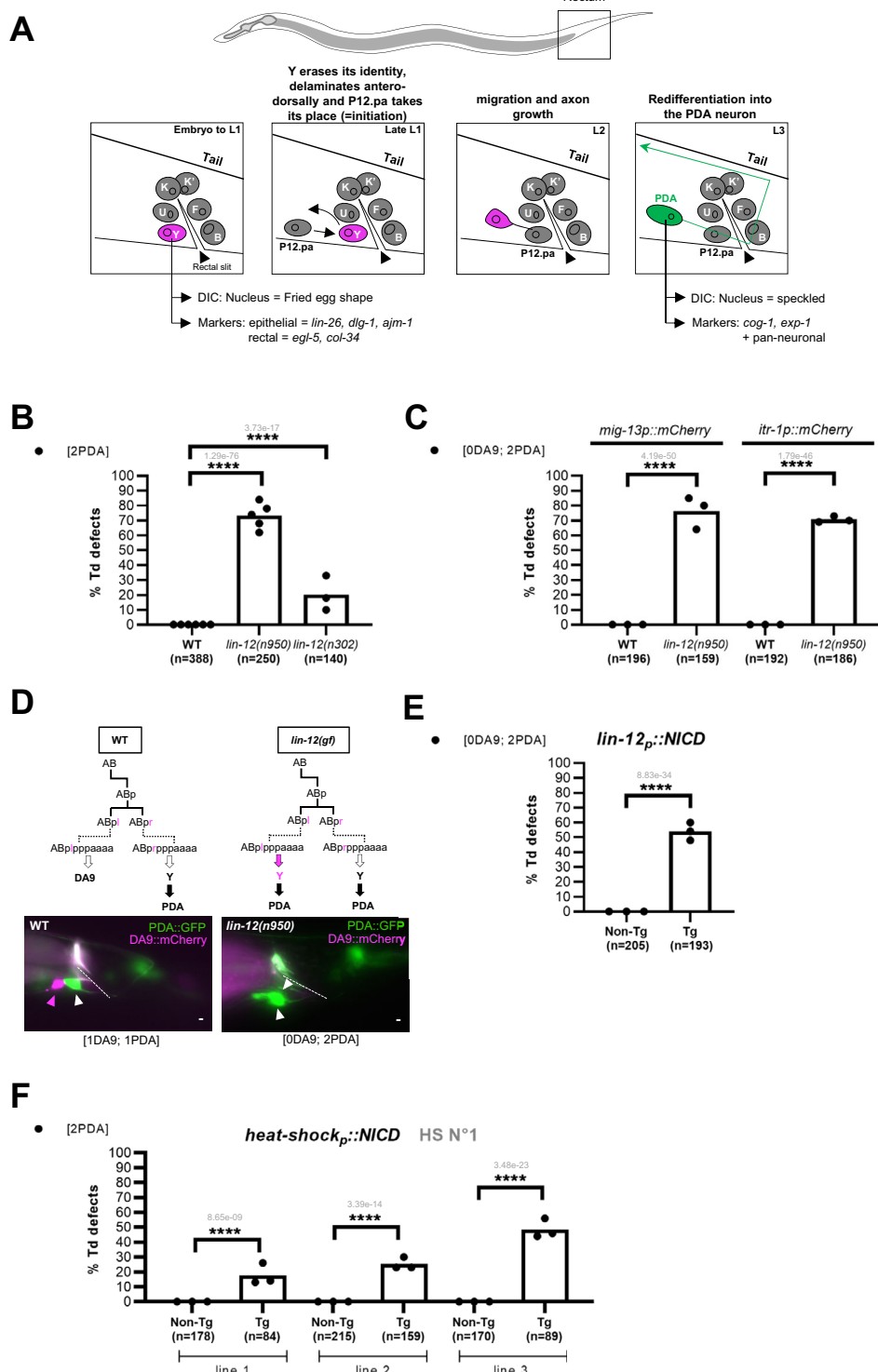

To confirm that *hlh-16* is required for the Td initiation as opposed to the earlier Y differentiation into a rectal cell, we examined if late overexpression of *hlh-16* would rescue the Td defect seen in *hlh-16(fp12)* mutant. Late *hlh-16* overexpression in the rectal cells, using the *col-34* promoter[32] (Fig. 2C), rescued the "0 PDA" mutant phenotype (Supplementary Fig. 1E). Thus, *hlh-16* is necessary for Y cell Td at the time of its initiation (and not at its birth), in accordance with recent findings by Rashid et al.[40]. In addition, combined with our findings that *hlh-16* is only expressed in the Y cell among the rectal cells (Fig. 3D), these results strongly suggest that *hlh-16* acts in a cell-autonomous manner to allow the initiation of Y Td.

**Both endogenous and *Notch(gf)*-induced Td require the same plasticity factors**

We have previously identified 5 other nuclear factors as being necessary for the initiation of Y Td: the members of the NODE complex (*ceh-6/Oct, sem-4/Sall, egl-27/Mta1*), *sox-2/Sox-2*, and the downstream factor *egl-5/Hox*[28,32]. Does the Td of the supernumerary Y cell in *lin-12^Notch(gf)* require these same plasticity factors as the endogenous Y-to-PDA Td? We thus examined if RNAi inactivation of these factors in a *lin-12^Notch(gf)* genetic background led to suppression of the "2 PDA" phenotype. Since *hlh-16* acts at the same step as the plasticity cassette factors necessary for the initiation of Y Td, we

**Fig. 1 | Ectopic Notch activity during early-embryogenesis results in an extra transdifferentiation event. A** Timeline of the Y-to-PDA Td. The rectum of the worm is composed of 6 specialised epithelial cells (Y, B, U, F, K and K') arranged in three rings. These cells, as well as DA9[prog], are born in the embryo around 290 min after the first cleavage[75] and are fully differentiated when the worm hatches. The Y cell keeps its epithelial identity until the end of the first larval stage, when Td is initiated. Y retracts from the rectum and migrates antero-dorsally, while it is replaced by P12.pa. Concomitantly, the Y identity is erased in a *sensu strictu* dedifferentiation step. Later in L2, re-differentiation into a motoneuron begins, to adopt the "PDA" final identity by the L3 stage. This sterotyped event is identifiable and predictable in all WT animals. **B** Quantification (in %) of [2 PDA] phenotype in WT, strong *lin-12*[Notch]*(n950)* and mild *lin-12*[Notch]*(n302)* gain-of-function alleles. **C** Quantification (in %) of [0 DA9; 2 PDA] in *lin-12*[Notch]*(n950)*. **D** Top panel: Lineage tree and fate of DA9[prog] and Y[prog] cells in WT, and *lin-12*[Notch]*(gf)* animals. Dotted line represents cell division not displayed in the scheme (after Sulston et al. 1983). Bottom panel**:** Pictures of the WT (left) and *lin-12*[Notch]*(n950)* [0 DA9; 2 PDA] (right) worms. White arrowheads: PDA neuron. Magenta arrowheads: DA9 neuron. Scale bar: 10 μM. **E** Quantification (in %) of the number of PDA and DA9 neurons found in a *lin-12*[Notch]*p*::NICD::*lin-12*[Notch]*3UTR′* transgenic line. **F** Quantification (in %) of worms exhibiting [2 PDA] in independent transgenic lines expressing the *hsp*::NICD construct, after a heat shock during early-embryogenesis (see HS 1, Fig. 2C). **B**–**F** n, total number of animals scored. Tg, transgenic animals. Non-Tg non-transgenic (control) siblings. *cog-1::gfp*, PDA marker. *mig-13p::mCherry* or *itr-1p::mCherry:* DA9 markers. %, proportion of worms showing the mutant phenotype over all worms scored (ie, penetrance). Data represent the mean of at least three biological replicates (dots represent the mean of each replicate). Two-tailed *P* value (in grey) is calculated using a Chi[2] test. ****$P < 0.0001$, ***$P < 0.001$, **$P < 0.01$, *$P < 0.05$.

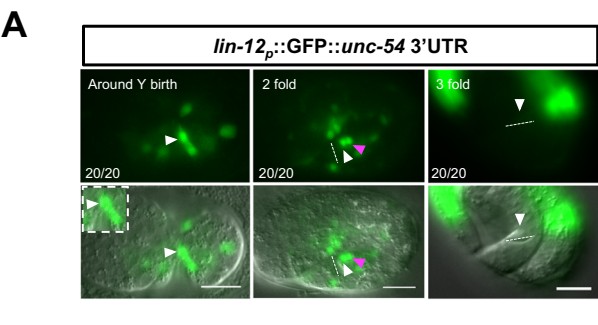

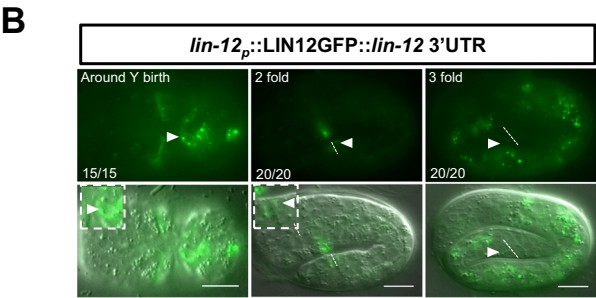

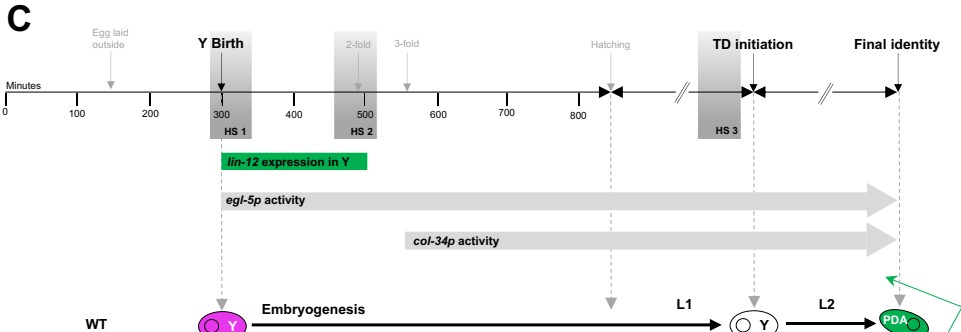

**Fig. 2 | *lin-12*[Notch] receptor is expressed from early to mid-embryogenesis.**
**A** Expression pattern and dynamics of *lin-12*[Notch]*p*::*gfp* (*rtEx727*). *lin-12*[Notch] is expressed in Y (arrowhead) during embryogenesis from its birth to the twofold stage. In the threefold stage, out-of-focus background GFP signal is due to expression of *myo-2*::*gfp*, the co-injection marker used to generate this transgenic line; no *lin-12*[Notch]*p*::*gfp* signal is detected in Y nor DA9 at this stage. **B** Expression pattern and dynamics of the LIN-12[Notch] protein (*arIs41*). LIN-12[Notch] is present in Y during embryogenesis from Y birth to the twofold stage (where a faint signal can still be detected). **A**, **B** Dotted line, rectal slit. Anterior is to the left and ventral to the bottom. Inserts represent blown-up rectal areas. White arrowheads: Y cell. Magenta arrowheads: DA9 neuron. Numbers represent the fraction of worms displaying this representative expression pattern over the total number of worms scored. *N* = 1. Scale bar: 10 μM. **C** Timing of expression of the *lin-12*, *egl-5* and *col-34* genes or drivers used, and of the heat-shocks (HS) 1, 2 and 3 with respect to the timeline of the different steps of the Y-to-PDA Td.

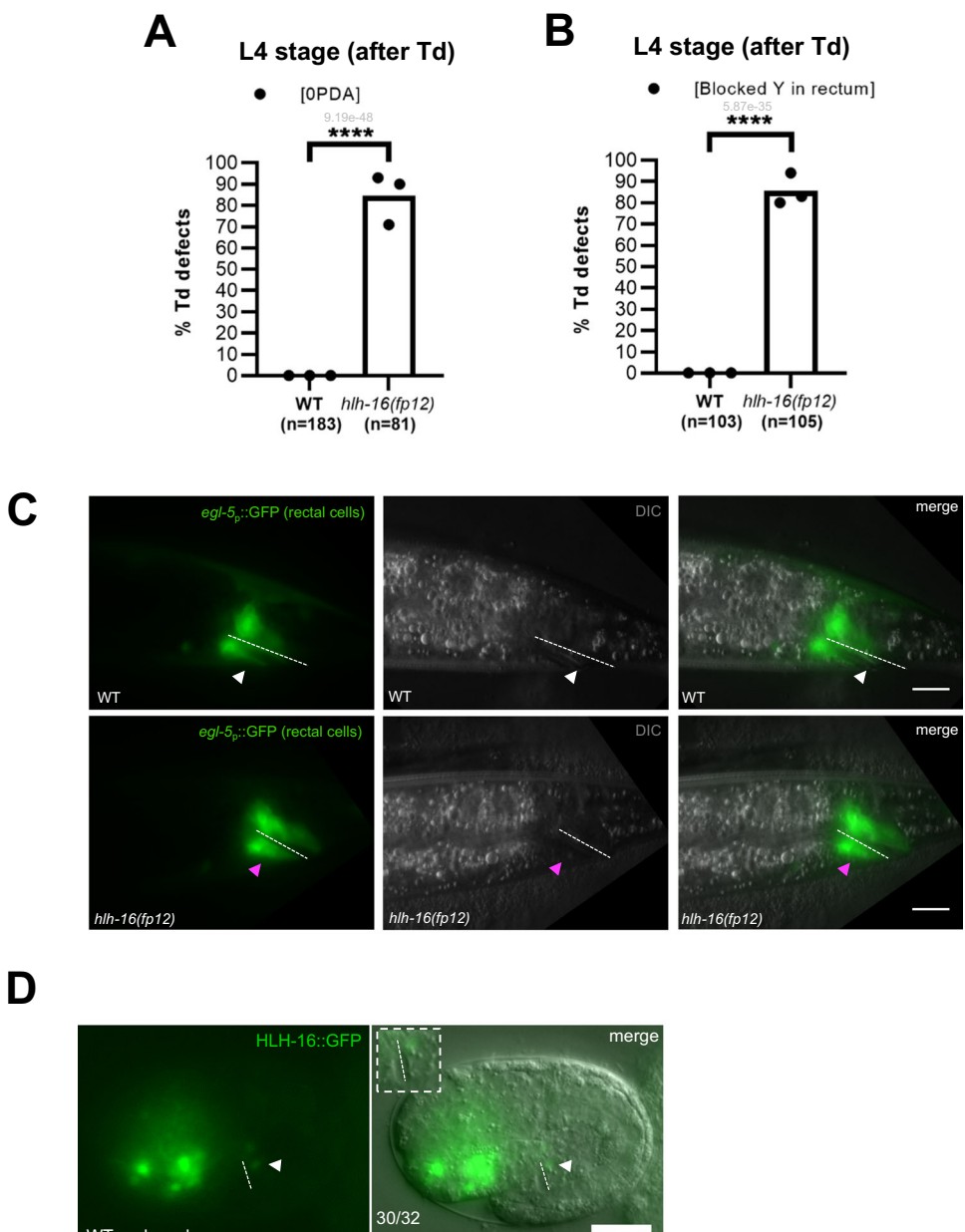

**Fig. 3 | _hlh-16_ is necessary for Y Td. A** Quantification (in %) of PDA presence at the L4 stage (after Td) in _hlh-16(fp12)_. _cog-1::gfp_, PDA marker. **B** Quantification (in %) of a persistent Y cell in the rectum at the L4 stage (after Td) in _hlh-16(fp12)_, assessed with _egl-5_ reporter. **A**, **B** _n_, total number of animals scored. Data represent the mean of replicates (dots: mean of each replicate). %, proportion of worms showing the mutant phenotype over all worms scored (i.e., penetrance). Two-tailed P value (in grey) is calculated using a Chi² test. ****_P_ < 0.0001. **C** Picture of the phenotypes represented in (**B**). Top: WT L4 animals; bottom: _hlh-16(fp12)_ L4 mutant. White arrowhead: absence of Y in WT. Magenta arrowhead: persistent Y cell in the rectum. Scale bar: 10 μM. **D** HLH-16::GFP is expressed in Y^prog but not in DA9^prog in the early WT embryo (1.5-fold). Insert represents blown-up rectal area. Dotted line, rectal slit. Numbers represent the fraction of worms displaying this representative expression pattern over the total number of worms scored. _N_ = 1. Scale bar: 10 μM.

examined if _hlh-16_ was also necessary for the _lin-12^Notch_-induced additional Td. Our results showed that these factors are necessary for both the additional and the endogenous Y-to-PDA Tds, as RNAi against these genes led to a significant reduction of the "2 PDA" phenotype and the appearance of a "0 PDA" phenotype (Fig. 4A, Supplementary Fig. 2A, B). In addition, a concomitant "0 DA9" phenotype is observed: DA9^prog is mis-specified into a Y cell in _lin-12^Notch(gf)_ mutant ("0 DA9"), that cannot transdifferentiate in absence of these factors ("0 PDA"). Using the same assay, we also found that _unc-3_, a factor we have shown to mediate the subsequent re-differentiation step[31] (Fig. 1A), appears to be necessary for the

formation of the supernumerary PDA, as expected if the same mechanisms are at play (Fig. 4B, Supplementary Fig. 2C).

Altogether, our results suggest that the Notch-induced ectopic Td event requires the same molecular mechanisms as the endogenous Y-to-PDA Td.

### HLH-16 and SEM-4, key factors for Td initiation, are downstream of Notch signalling

We next asked if the Notch pathway conferred competence to transdifferentiate by promoting plasticity factors expression. To this aim, we compared expression of the plasticity factors (NODE complex (_ceh-_

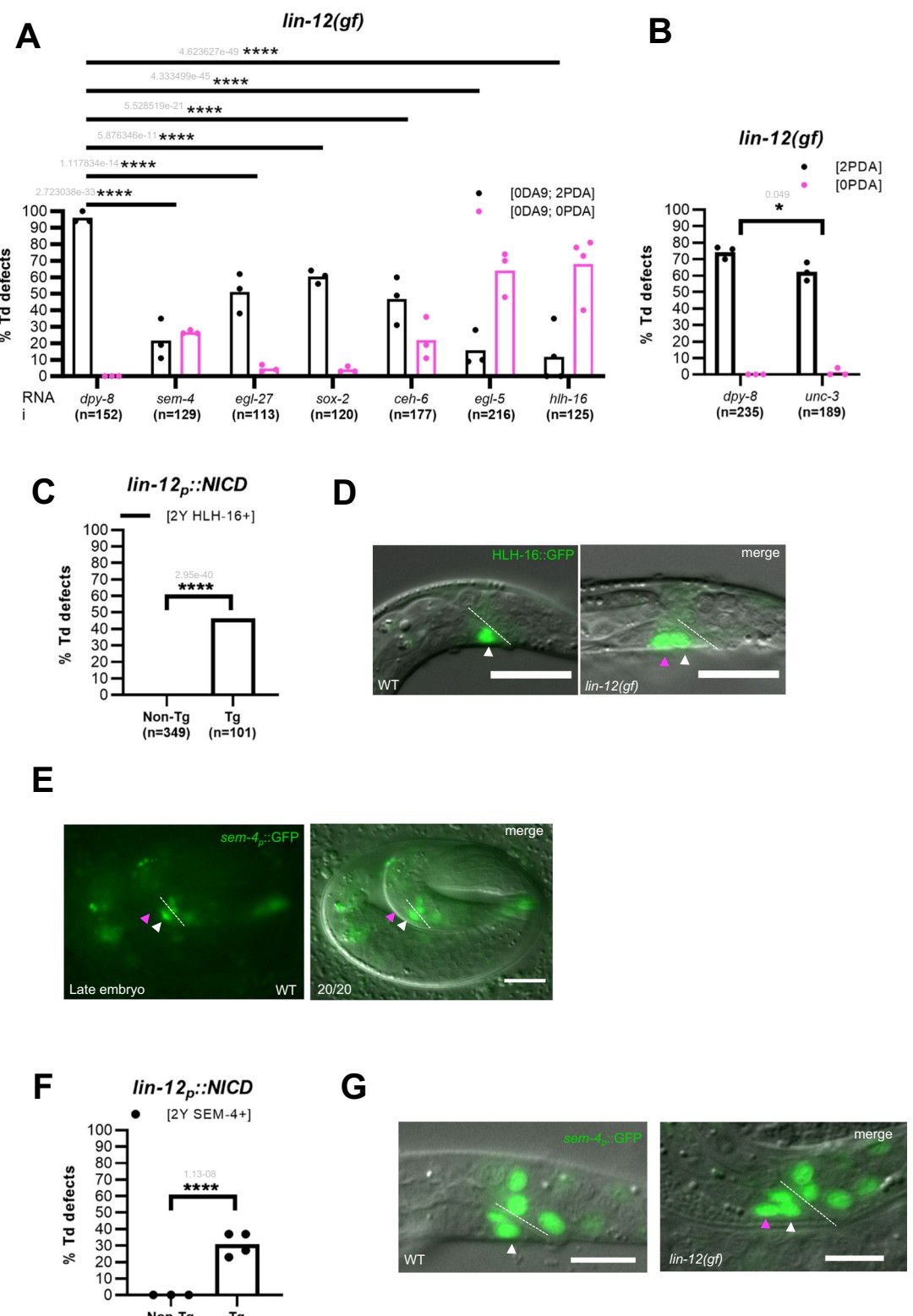

6/Oct, sem-4/Sall, egl-27/Mta1), sox-2/Sox-2, egl-5/Hox and hlh-16/Olig) in WT and worms carrying the lin-12$^{Notch}$p::NICD transgene (active form of the lin-12$^{Notch}$ receptor).

We found two types of patterns: i) genes endogenously expressed in both the Y$^{prog}$ and DA9$^{prog}$ cells in WT animals since their birth. This is the case for the egl-5, sox-2 and ceh-6 genes (Supplementary Fig. 2D–F). Since these genes are already expressed in the DA9$^{prog}$ cell since its birth in the absence of ectopic Notch, we concluded that these genes

are not downstream of Notch. As egl-5, sox-2 and ceh-6 genes are necessary for Y Td, but not Y identity[28,34], these genes could either contribute to a Notch-permissive context, or allow the Y cell Td independently of Notch.

And, ii) genes that, besides being naturally expressed in the Y$^{prog}$ cell, are only observed in the extra Y cell coming from the mis-specification of DA9$^{prog}$ after receiving ectopic lin-12$^{Notch}$ signalling. This is the case for the hlh-16 and sem-4 genes (see the Y cell and a second

**Fig. 4 | Additional Td requires endogenous Td factors, unveiling *hlh-16* and *sem-4* as downstream of Notch. A** Plasticity factors necessary for the WT Y-to-PDA Td are also required for the extra Td ([2 PDA]) in *lin-12*[Notch](gf). The [2 PDA] (in %) is significantly suppressed when the plasticity factors (*sem-4, egl-27, ceh-6, sox-2, egl-5* and *hlh-16*) are knocked-down by RNAi in *lin-12*[Notch](n950) and a concommittant 0 PDA phenotype is observed (*dpy-8*, negative control). See also Supplementary Fig. 2A, B. **B** Significant suppression of the *lin-12*[Notch](gf) [2 PDA] (in %) when *unc-3* is knocked down by RNAi in *lin-12*[Notch](n950). *dpy-8*, negative control. Of note, *unc-3*(RNAi) is poorly efficient and leads to a very low penetrance defect. **A, B** *cog-1::gfp*, PDA marker. *itr-1p::mCherry*, DA9 marker. Quantification of the number of HLH-16 + Y (**C**) and SEM-4 + Y (**F**) cells in the rectum (in %) of *lin-12*[Notch]*p::NICD* L1 animals.

HLH-16::GFP (**D**) and *sem-4p::GFP* (**G**) are expressed in Y WT L1 larva (white arrowhead). As quantified in (**C, F**), 2 Y cells expressing HLH-16::GFP (magenta and white arrowheads, **D**) and *sem-4p::GFP* (**G**) are found in *lin-12*[Notch]*p::NICD*. **E** *sem-4p::gfp* is expressed in Y[prog] but not in DA9[prog] in the late WT embryo (threefold). *N* = 1. Numbers in bottom left corner represent the fraction of worms displaying this representative expression pattern over the total number of worms scored. **A–C, F)** *n*, total number of animals scored. %, penetrance of the mutant phenotype shown. Two-tailed P value (in grey) is calculated using a Chi[2] test. Tg: transgenic; non-Tg: non-transgenic (control) siblings. ****P < 0.0001, ***P < 0.001, **P < 0.01, *P < 0.05. **D, E, G** Anterior to the left, and ventral at bottom. Dotted line: rectal slit. Scale bar: 10 µM.

cell persistently expressing *hlh-16* or *sem-4* in Figs. 3D, 4C, D, E–G). A second Y cell is observed in *lin-12*[Notch](gf) mutant worms with the same penetrance as the "2 PDA" phenotype. Consistently, *hlh-16* or *sem-4* expression starts in the Y cell quite some time after its birth (Figs. 3D–4E). Indeed, *sem-4* is observed in Y rather late (when the embryo is fully elongated, Fig. 4E), while *hlh-16* starts to be expressed in comma stage embryos (Fig. 3D), and their expression is maintained in the Y cell during the L1 stage (Figs. 4D, G). It thus appears that in endogenous Y[prog] and in DA9[prog] experiencing ectopic Notch signalling, expression of *hlh-16* and *sem-4* is triggered downstream of Notch signalling. Of note, expression of *hlh-16* and *sem-4* persists until Td initiation, while LIN-12[Notch] expression ceases during embryogenesis (twofold, Fig. 2A–C), suggesting that Notch signalling is necessary to promote their initial expression but that expression maintenance is most likely Notch-independent.

Therefore, the Notch pathway enables the Y cell competence to transdifferentiate by promoting plasticity factors expression such as *hlh-16* and *sem-4*.

## Ectopic Notch signalling can both promote and block Y-to-PDA Td

Interestingly, when broadly expressing NICD under the hs promoter in the whole embryo, we found that these animals had two, and not many, PDA neurons (Fig. 1F). This suggested that, at this Notch signalling dose, some cells like DA9[prog], are more prone to be transformed into a Y cell that is competent to Td. In particular, the other rectal cells, born at the same time as Y[prog], do not appear to change their identity nor their typical fried-egg-shaped nuclear morphology. We next tested if using a promoter driving robust and maybe longer NICD expression specifically in these cells could allow their conversion into supernumerary PDA neurons. To this end, the constitutively active NICD was expressed under a rectal promoter, *egl-5p*[28], which expression starts shortly after the rectal cells' birth (Fig. 2C). The identity of the rectal cells was monitored at the L4 stage using both appearance under DIC microscopy and rectal markers (*egl-5, lin-48* and *mab-9*, Fig. 5A–C). We found that continuous Notch signalling in these cells did not result in a switch in their identity, and no extra PDA neurons were observed (Fig. 5D). Therefore, by contrast to DA9[prog], these other rectal cells represent a refractory context for the Notch-promoted Td, highlighting the importance of a particular cellular context for the efficiency of a given reprogramming cue. Thus, a permissive intrinsic context is necessary for Notch signalling to induce a cell with the capacity to transdifferentiate.

A closer look at the worms carrying the transgene *egl-5p::NICD* showed to our surprise that a large number of them did not have a PDA neuron ("0 PDA", Fig. 5D). Indeed, we expected most worms to exhibit 1 or 2 PDA neurons, based on our experiments overexpressing NICD using the *lin-12* promoter or after an early heat shock (Fig. 1E, F), and knowing that *egl-5* promoter allows a similar onset of expression in the Y cell. In addition, the rectal Y cell was still present in the rectum of most L4 stage worms expressing *egl-5p::NICD* (Fig. 5A, Supplementary Fig. 3A, B), suggesting that this treatment resulted in a block of the endogenous Td of the Y cell. We therefore analysed how ectopic Notch signalling can both promote formation of ectopic PDAs or lead to a Td

blockage ("0 PDA" phenotype), hereafter called "dichotomic effect". Could perdurance of NICD expression lead to a failure of endogenous Y-to-PDA? The *egl-5* promoter used to express NICD in the rectal cells is active from Y birth to the initiation of Td (Fig. 2C). However, in WT worms, Notch receptor expression stops after the twofold stage (Fig. 2A–C). This suggests that the "No Td" ("0 PDA") phenotype obtained when expressing NICD under the *egl-5* promoter could be a consequence of a prolonged Notch signalling in the Y cell. To determine if the timing of ectopic Notch signalling was crucial for this dichotomous effect on Y Td, we expressed NICD in the rectal cells later, starting from the late embryonic threefold stage, using the *col-34* promoter[32] (Fig. 2C). This also resulted in a strongly penetrant "0 PDA" phenotype (Fig. 5E), suggesting that it is either a continuous or a late Notch signalling that was detrimental to Y Td. We next tested the impact of Notch signalling at specific time points (Fig. 2C) by providing a pulse of NICD under the hs promoter: i) at the embryonic twofold stage, when endogenous *lin-12*[Notch] expression disappears (see HS 2, Fig. 5F), and ii) during the L1 larval stage, just before the initiation of the Td process in WT worms (see HS 3, Fig. 5G). While the expression of NICD at the twofold stage resulted in a few instances of the "0 PDA" phenotype (and also in few "2 PDA", Fig. 5F), expression at the time of Td initiation resulted in a significant number of "0 PDA" instances, and an absence of the "2 PDA" phenotype (Fig. 5G). These results suggest that: i) the Notch pathway could not promote the conversion of DA9[prog] into an extra competent Y cell anymore after the twofold stage (no "2 PDA" phenotype); ii) the closer the Y cell is to Td initiation when it encounters ectopic Notch pathway activation, the stronger the "0 PDA" phenotype is. Thus, the embryonic twofold stage represents a critical checkpoint for Notch signalling to be interpreted as a pro- or anti-Td signal (mix of "0 PDA" and "2 PDA" phenotypes). In addition, while the different drivers used may lead to variable levels of Notch signalling, our data suggest that the critical parameter is encountering Notch signalling at the Td initiation time, independent of Notch levels. Altogether, these results suggest that the time window during which the Y cell receives a Notch signal is key for its capacity to transdifferentiate; seeing Notch signalling, even transiently, at the time of the Td initiation blocks the identity conversion, points further developed in the discussion.

We then examined if late, ectopic Notch activation blocked Y Td cell-autonomously, or if it was due to Notch activation in (an)other cell(s). With a particular focus on the rectal cells, we drove the expression of NICD using various promoters active in subsets of them and examined the presence of PDA. First, we used the *lin-48* promoter, which is only active in the F, U, K and K' rectal cells[41,42] (Fig. 6A). Such partial rectal expression does not result in a " 0 PDA" phenotype (Fig. 6B), implying that an activated Notch signal in F, U, K and K' is not responsible for the Td defect observed with the *egl-5p::NICD* transgene. This suggested that Notch signalling acts either in the Y or the B cell. To discriminate between the B and Y cells, we used the mosaic expression our *egl-5* transgene. We focused on the expression of the transgene in the B cell and found no correlation between NICD expression in the B cell and the appearance of a "0 PDA" phenotype (Fig. 6C). We also used the *egl-20p* promoter,

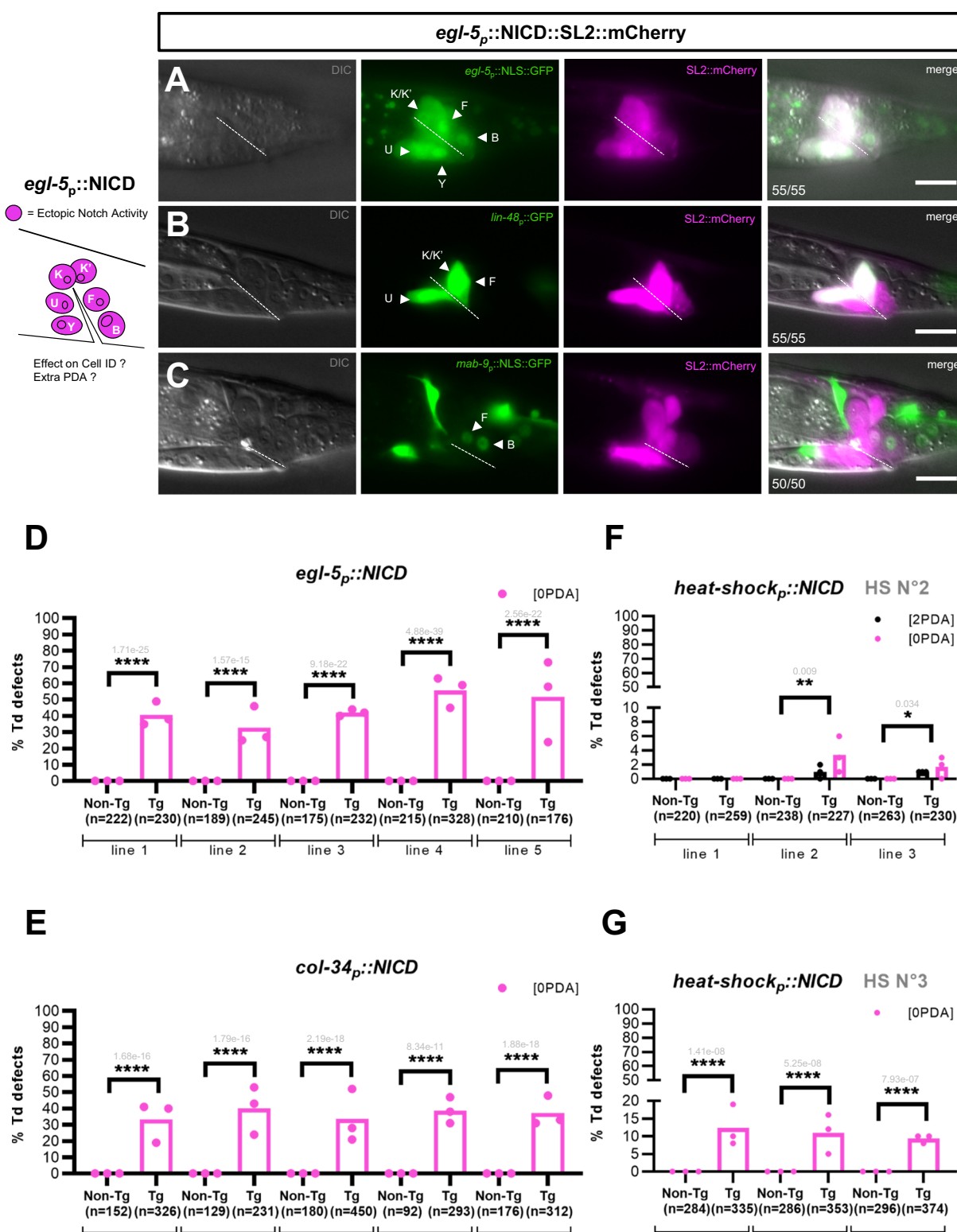

previously described as being expressed in all rectal cells except Y[43]. We however observed, similarly to others[44], that *egl-20* is in fact expressed in Y with a high degree of mosaicism (Supplementary Fig. 3C-D). Nevertheless, most of the worms displaying a "0 PDA" phenotype did not express NICD in the B cell (Fig. 6D-Supplementary Fig. 3C–E), further suggesting that it is Notch activation in the Y cell that blocks its Td.

Altogether, these data suggest that Y Td is blocked by Notch over-activation in a cell-autonomous manner.

## Late Notch signalling blocks Td initiation by over-imposing a rectal identity

Why does a late Notch signal lead to an impairment of Y Td? To address this question, we examined the specific step at which Y Td is blocked, by analysing the morphology and position of the Y cell, plus the markers it expresses when it receives a late Notch signal. In WT worms, after Td has taken place, only two hypodermal cells are present in the anterior part of the rectum (P12.pa and U, Fig. 6E). Interestingly, in worms overexpressing the NICD transgene, three hypodermal-like

**Fig. 5 | Ectopic Notch signal in rectal cells is not sufficient to form additional PDAs.** Left, experimental approach: overexpression of the NICD construct in all the rectal cells using the *egl-5* promoter. Representative pictures of the specified marker expression in a L4 worm (**A**, **C**) or a L3 worm (**B**), showing that the rectal identity of the B, U, F, K and K' rectal cells is not affected by NICD over-expression. SL2::mCherry shows NICD expression in the nucleus of rectal cells (see also Supplementary Fig. 3). No constipation is observed, showing that these rectal cells are fully functional. Bottom left corner, number of animals showing the represented phenotype over the total number of animals scored. N = 2. Anterior is to the left and ventral to the bottom. Arrowheads show the position of the designated cells. Dotted line, rectal slit. Scale bar: 10 μM. **D**–**E** High penetrance of Td defect [0 PDA] (in %) is found in transgenic ("Tg") worms expressing *egl-5p::NICD::SL2mCherry* (**D**) or *col-34p::NICDGFP* (**E**) in five independent transgenic lines. No extra Td [2 PDA] was observed. Non-transgenic ("Non-Tg") control siblings are WT as observed with *cog-1::gfp* reporter. See Fig. 2C for the timing of *egl-5* and *col-34* expression. **F**, **G** Quantification (in %) of worms exhibiting an extra Td [2 PDA] or no Td [0 PDA] phenotype after a heat shock during (**F**) mid-embryogenesis (HS 2, Fig. 2C) and (**G**) before Td initiation in late L1 stage (HS 3, Fig. 2C), in three independent transgenic lines expressing the *hsp::NICD* construct, as assessed with *cog-1::gfp* reporter. **D**–**G** n, total number of animals scored. %, penetrance of the mutant phenotype shown. Data represent the mean of 3 replicates; dots: mean of each replicate. Two-tailed P value, indicated in grey, is calculated using a Chi² test. ****P. < 0.0001, ***P. < 0.001, **P. < 0.01, *P. < 0.05.

cells are present in the anterior part of the rectum, including the Y cell (Fig. 6E). This mimics the phenotype of mutants where Td is blocked before initiation and the Y cell remains rectal at its original position[28,30-33]. In such mutants, the Y cell keeps its epithelial identity and does not express any neuronal markers. To investigate whether Y remains epithelial when exposed to late Notch signalling, we examined epithelial or rectal-specific markers in worms expressing *egl-5p::NICD*. We used two epithelial junction markers, *AJM-1::GFP*, *DLG-1::GFP*, a rectal marker, *egl-26p::GFP*, and an epithelial differentiation marker, *lin-26p::NLS-GFP*[28] (Fig. 6F-Supplementary Fig. 4A–D). By contrast to the WT pattern, each of these markers remained expressed in the Y cell past the L2 stage, indicating that late Notch activity in Y leads to an early block of the Td process, with no retraction of Y from the rectum (Fig. 1A) and maintenance of its rectal and epithelial identity. To rule out the possibility that Y is blocked in a mixed identity state in which both epithelial and neuronal markers are expressed—something that never happens during the wild-type process[31]—we looked for the potential expression of a second PDA marker, *exp-1p::GFP*, and three pan-neuronal markers, *tag-168p::GFP*, *unc-33p::GFP* and *unc-119p::GFP*, in the persistent Y. None of these were observed in the blocked Y (Fig. 6F, Supplementary Fig. 4E–H). Collectively, our data show that late over-activation of the Notch pathway in the Y cell immediately prior to Td blocks its initiation by re-enforcing a rectal identity.

## Downregulation of Notch in the Y cell occurs at the transcriptional level

Our discovery, that the time window during which the Y cell receives a Notch signal is crucial for proper Td, led us to examine how endogenous Notch activation is regulated. We first sought to identify the ligand(s) that activate it. Functional (3) or predicted (12) ligands and co-ligands exist in *C. elegans*[21,38,45] and were tested for a role in Y-to-PDA using available mutants and RNAi in sensitised backgrounds (Supplementary Fig. 5A, B). We found that the loss of *apx-1* or *lag-2* led to a Y-to-PDA defective Td ("0 PDA"), consistent with their role as ligands (Fig. 7A, B), while loss of any of the other potential ligands and co-ligands did not impair Td (Supplementary Fig. 5B). Among the various strategies to enable fine control of Notch signalling, ligand unavailability[46] could result in switching off *lin-12^Notch* receptor activity. Examination of the *lag-2* expression pattern showed that it is expressed from Y birth until the beginning of Td in the rectal B cell, which directly contacts and forms adherens junctions with Y[28] (Fig. 7C). *apx-1* is also expressed in four cells close to the rectum at Y birth. After the embryonic twofold stage, two of these cells maintain this expression and remain in close contact with the rectum until at least beginning of Td (Fig. 7D). Thus, these two ligands are expressed around the Y cell, and their expression persists from its birth until at least the initiation of Td, a profile suggesting that the disappearance of the Notch signalling in Y after the embryonic twofold stage is not due to the unavailability of the ligands. We also note that absence of ligand expression in the DA9^prog and Y^prog themselves, together with previous laser ablations[29], indicate that a Notch inductive signal is involved in potentiating Y with the competence to Td.

Are the ligands still functional after LIN-12^Notch has disappeared from Y? If yes, they should be able to activate an ectopic full-length LIN-12^Notch receptor re-expressed in Y late, after the endogenous one had disappeared (using a *col-34* promoter), and thus block Td. We indeed observed a No Td [0 PDA] defect, while non-transgenic controls were WT (Fig. 7E, Supplementary Fig. 6A). These results show that the ectopic LIN-12^Notch receptor has been activated by the surrounding ligands, confirming that its ligands are indeed not only available but also functional until initiation of Td. As negative controls, we used forms of the LIN-12^Notch receptor that cannot be activated: a receptor-dead form [*lin-12^Notch* cDNA bearing the premature stop codon in the extracellular EGF repeats as found in the null allele *lin-12^Notch(n914)*[47]] and a LIN-12^Notch receptor unable to signal [*lin-12^Notch* cDNA bearing a deletion of the seven ankyrin repeats of the intracellular part of LIN-12^Notch receptor[47]]. No inhibition of Y-to-PDA Td were observed in transgenic animals expressing these two constructs in Y late, as expected (Fig. 7E, Supplementary Fig. 6B, C), suggesting no ectopic activation of the Notch pathway. As a positive control, a full-length mutant receptor that is constitutively active was expressed [*lin-12^Notch* cDNA containing the mutation found in the *gf lin-12^Notch(n137)* allele, which has been shown to induce ligand-independent receptor activation[45,48,49]]. Transgenic animals expressing this construct exhibited a penetrant "0 PDA" phenotype (Fig. 7E, Supplementary Fig. 6D), validating our conclusions.

Several lines of evidence point to a regulation of *lin-12^Notch* expression over time in the Y cell: (i) the LIN-12^Notch receptor disappears from the Y cell after the twofold stage (Fig. 2B) (ii) the transcriptional *lin-12^Notch* reporter showed the same expression dynamics (Fig. 2A); since the transcriptional reporter lacks the *lin-12^Notch* 3'UTR, these data suggested a transcriptional - rather than post-transcriptional - regulation of *lin-12^Notch*[36,37] (Supplementary Fig. 1B). To test if the promoter of *lin-12^Notch* is the main effector of LIN-12^Notch activity regulation, we re-examined our results expressing the NICD fragment under control of the *lin-12^Notch* promoter (Fig. 1E). If *lin-12^Notch* promoter is sufficient to recapitulate the brief WT pulse of Notch expression in the embryo, NICD expression under *lin-12^Notch* promoter should lead to activation of the Notch pathway in Y^prog and DA9^prog, and induce the appearance of an additional Td [2 PDA]. If not, NICD would be expressed after the embryonic twofold stage in Y, leading to a block of Td [0 PDA]. Worms carrying this transgene displayed a "2 PDA" phenotype while the controls were all WT ("1 PDA") (Figs. 1E, 7F, Supplementary Fig. 6E). These results show that down-regulation of the Notch signalling in the Y cell is achieved through transcriptional modulation during embryogenesis, a crucial checkpoint to allow the subsequent initiation of Y Td. We further identified in the *lin-12^Notch* promoter two highly conserved regions (R1 and R2) between different *Caenorhabditis* species (Fig. 7G), with deletion of the R2 region completely abolishing the activity of the transgene (Fig. 7F, G, Supplementary Fig. 6F, G). In conclusion, we have found that endogenous Notch expression is tightly regulated at the transcriptional level during embryogenesis, and a conserved region in its promoter is necessary for its transient expression during Y Td.

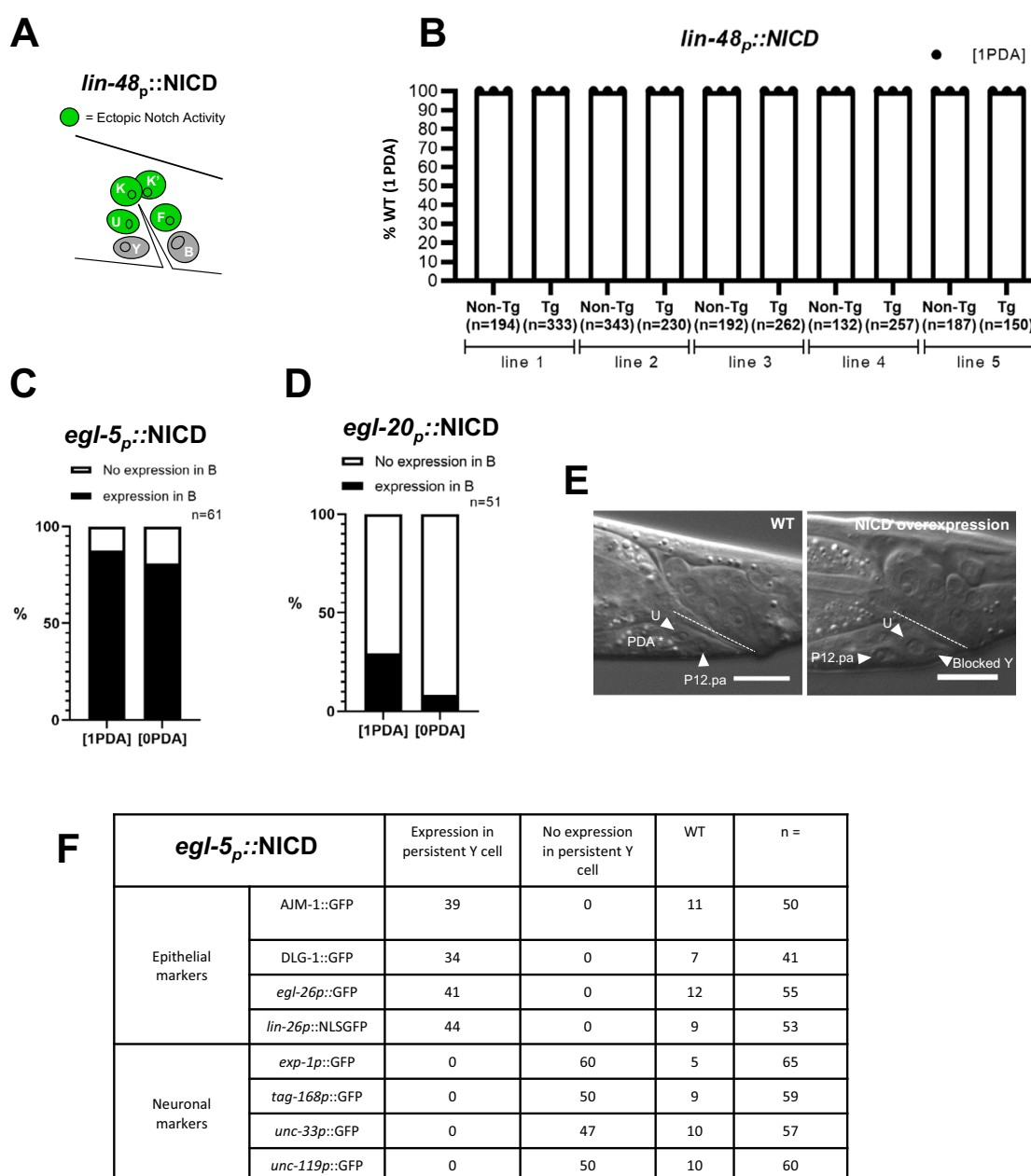

**Fig. 6 | A late Notch signal cell-autonomously blocks initiation of Td.**
**A** Experimental approach: overexpression of the NICD construct in a subset of the rectal cells using the *lin-48* promoter. **B** No Td defects ([0 PDA], in %) is observed when NICD (*lin-48p::NICD::SL2mCherry)* is expressed in the F, U, K and K' rectal cells in five independent transgenic lines as assessed using *cog-1::gfp* reporter. Tg transgenic worms, Non-Tg non-transgenic WT siblings, n total number of worms scored. Data represent the mean of 3 replicates; dots: mean of each replicate. A similar proportion (**C**) or even a higher proportion (**D**) of animals (in %) exhibit a Td defect [0 PDA] when NICD is not expressed (white bar) in the B cell using a mosaic *egl-5* driver (**C**) or a mosaic *egl-20* driver (**D**), revealing no correlation between Td defect and expression in the B cell; *n*, total number of animals scored = 61 (**C**) and 51 (**D**). **E** Representative DIC picture of the persistent Y cell, and thus ([0 PDA]) as quantified in (**C**), in L4

transgenic animals expressing ectopic activated Notch in the rectal cells (*egl-5p::NICD)*. Left, WT worm where Y has turned into a PDA (star). Right, L4 worm with ectopic Notch activation in Y. The Y cell (arrowhead) is blocked before Td initiation and found at its original position in the rectum; the rectal U and P12.pa cells (arrowheads) are also visible. Dotted line, rectal slit. Anterior is right and ventral is to the bottom. Scale bar: 10 μM. **F** The persistent Y cell remains rectal. Epithelial and neuronal markers in the persistent Y cell (expressed in number of animals) in worms expressing an integrated NICD construct under the *egl-5* promoter (*egl-5p::NICD)*. When Y Td is blocked, a persistent Y cell is found at its original position (160/200 animals, *N* = 4). This Y cell remains rectal. See also Supplementary Fig. 4. n, total number of animals scored.

## Notch levels do not regulate the timing of Y-to-PDA Td

The ability of ectopic Notch activity to promote or block the Y-to-PDA Td could suggest an involvement of the Notch pathway in regulating the timing of this reprogramming event. We were particularly intrigued by the suggestion that Notch levels modify the temporality of Y-to-PDA Td, as recently proposed in Rashid et al.[40]. They reported premature occurrences of Y-to-PDA Td at the onset of the L1 stage

instead of the beginning of the L2 stage in WT, in a specific *lf* allele of *lin-12*[Notch], *n676n930*, that has been described as a weak *gf* allele in some particular contexts at 15 °C[50]. Indeed, using 15 °C as a putative *gf* temperature, they reported that the Y cell already exhibited an axon at the beginning of the L1 stage, while still expressing genes observed in the rectal Y cell (*ngn-1* and *hlh-16*), leading to the designation of this cell as a "precocious PDA". However, this putative "precocious PDA"

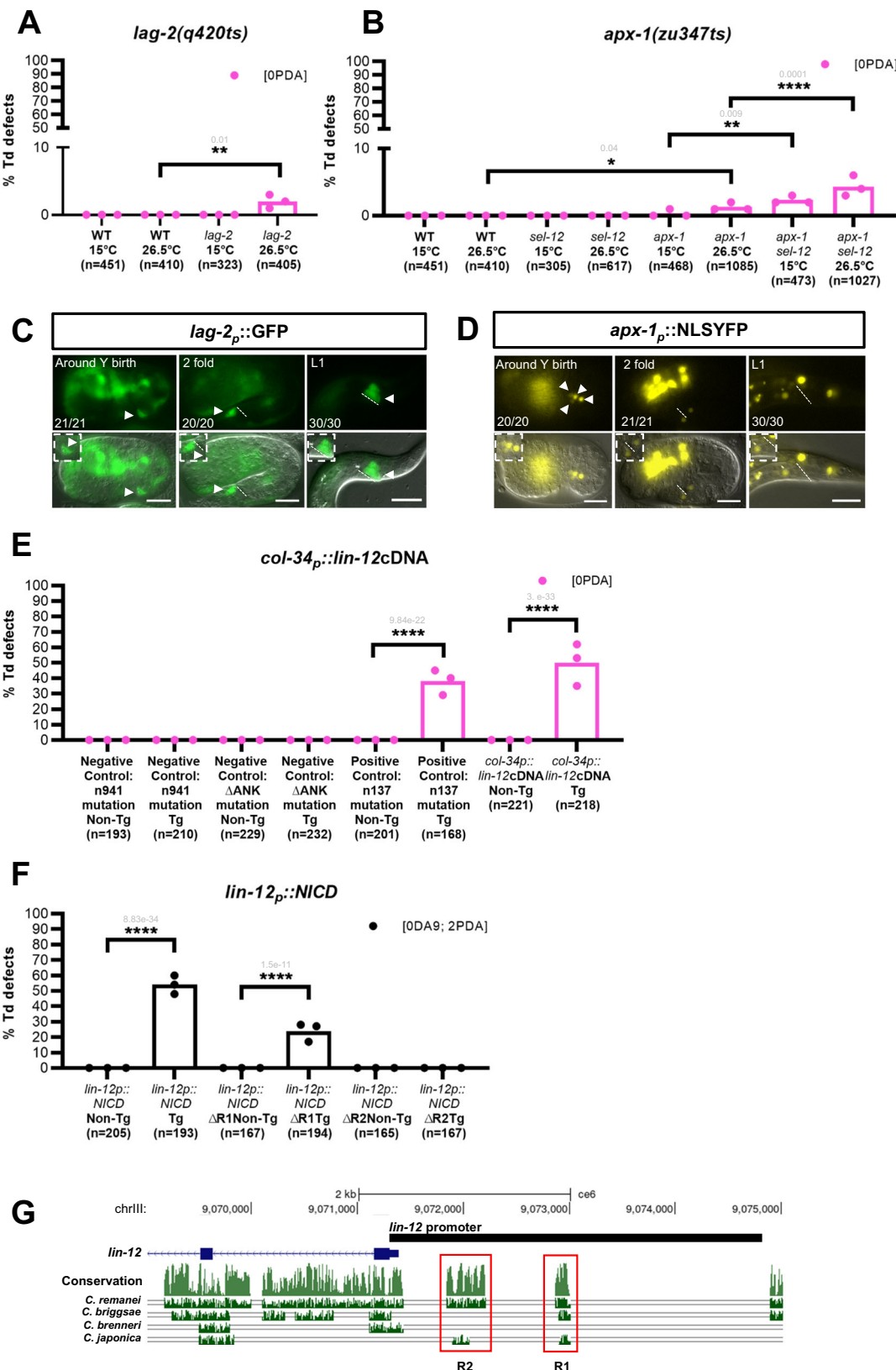

then failed to develop into a complete PDA, as suggested by the absence of PDA markers expression in older L4 animals[40]. Given their conclusion that a weak *lin-12^Notch gf* activity appeared to provoke an earlier Y cell reprogramming, they posited that the level of Notch signalling regulates the timing of the Y-to-PDA Td.

However, we had previously reported that this specific *lin-12^Notch*(*n676n930*) allele behaves as a *lf* at 15 °C in the Y cell, and that no Y cell, hence 0 PDA, are made[28]. Indeed, besides its role in the Y cell's competence to transdifferentiate[28], the Notch pathway is implicated in cell-fate decision between a DA9 neuron and a Y rectal cell

**Fig. 7 | Regulation of Notch signal duration is achieved through transcriptional regulation. A–D** *lag-2* and *apx-1* act as ligands for activation of Notch in the Y cell. Permissive temperature, 15 °C; restrictive temperature, 26.5 °C. **A)** Quantification (in %) of [0 PDA] phenotype in *lag-2(q420ts)* mutant. **B** Quantification (in %) of [0 PDA] in single *apx-1(zu347ts)* and double *apx-1(zu347ts); sel-12(ar171)* mutants. Notch pathway mutant *sel-12(ar171)* does not exhibit Td defect and is thus used as a sensitised background, see also Supplementary Fig. 5A. **C** *lag-2* is expressed in B (arrowheads) from Y birth until Td initiation. *N* = 1. Scale bar: 10 μM. **D** *apx-1* is expressed in four cells (arrowheads) close to the rectum during embryogenesis. **C**, **D** Dotted line: rectal slit. Inserts represent blown-up rectal areas. Numbers represent the fraction of worms displaying this representative expression pattern over the total number of worms scored. *N* = 1. Scale bar: 10 μM. **E** Ligands are still available and functional when *lin-12^Notch* is down-regulated. Last 2 bars, proportion of [0 PDA] worms carrying a *col-34p::lin-12^Notch(WT)cDNA* transgene. Negative control transgenic lines: inactive LIN-12^Notch receptors driven by *col-34p* [*col-34p::lin-12^Notch cDNA(n941)* and *col-34p::lin-12^Notch cDNA(ΔANK)*]. Positive control:

constitutively active full-length receptor driven by *col-34p* [*col-34p::lin-12^Notch cDNA(n137)*]. **F** Quantification (in %) of the number of PDA and DA9 neurons found in transgenic lines expressing NICD under different *lin-12^Notch* promoter variants. First 2 bars: *lin-12^Notch p::NICD* transgenic line (same data as Fig. 1E). Middle 2 bars: *lin-12^Notch p(ΔR1)::NICD* line. Right-most 2 bars: *lin-12^Notch p(ΔR2)::NICDGFP* line. **G** The *lin-12^Notch* promoter region (black rectangle) used to drive NICD expression. Light green and green tracks represent the overall conservation of the DNA sequence between different *Caenorhabditis* species and *C. elegans* respectively (adapted from UCSC genome browser). The red boxes identify two conserved regions (R1, R2) of the *lin-12^Notch* promoter further tested. **A, B, E, F)** PDA marker: *cog-1::gfp*. DA9 marker: *itr-1p::mCherry*. Tg transgenic worms, non-Tg non-transgenic siblings. n total number of animals scored. %, penetrance of the mutant phenotype shown. Data represent the mean of replicates; dots: mean of each replicate. Two-tailed P value, indicated in grey, is calculated using a Chi² test. ****P < 0.0001, ***P < 0.001, **P < 0.01, *P < 0.05. Additional lines are depicted in Supplementary Fig. 6A–G.

fate[17]. Specifically, Notch signalling has been shown to be necessary and sufficient for the Y cell fate[17]. In the presence of high Notch levels Y^prog becomes rectal while no or low Notch levels lead to mis-specification as a DA9 neuron; both fates appear independent of each other, as shown by ablation experiments[51]. Consequently, in Notch *lf* alleles, such as *lin-12^Notch(n676n930)* at 25 °C and at 15 °C, Y^prog is mis-specified as an extra DA9 neuron instead of a Y cell[28]. We thus re-examined this issue. Consistent with our previous report, we found that growing *lin-12^Notch(n676n930)* at 15 °C results in an absence of PDA, as would be expected for *lf* activity (Fig. 8A). Additionally, we never observed the distinct phenotype of *lin-12^Notch(gf)* alleles ("2 PDA" and an absence of DA9, Fig. 1B–E) in *lin-12^Notch(n676n930)* raised at 15 °C. We confirmed by sequencing the presence of the *lin-12^Notch (n676n930)* mutations in our strain, excluding a *lin-12^Notch* genetic issue as the basis for our diverging results from Rashid et al.[40].

To re-examine the possibility that we might have overlooked the presence of a "precocious PDA", we investigated the identity of the Y cell in 15 °C *lin-12^Notch(n676n930)* using a battery of markers. These experiments were conducted using the same protocol and at the precise time point (newly hatched L1) at which Rashid et al.[40] observed the appearance of a precocious PDA, and we additionally analysed the time of WT Td initiation (23.5 h post hatching at 15 °C). We first examined two different rectal markers, *egl-5* and *hlh-16*, and never observed a Y cell presenting an axon (purported "precocious PDA") (Fig. 8B, C). Instead, at both time points, an absence of the Y cell was observed in most of the worms (Fig. 8B–D). Since the number of DA9 neurons made in *lin-12^Notch(n676n930)* was not reported by Rashid et al., we also examined it using the DA9 cell marker *mig-13*[52]. Most *lin-12^Notch(n676n930)* worms raised at 15 °C and 25 °C exhibited 2 DA9 neurons, together with a concomitant absence of the Y cell (Fig. 8A, C, D), confirming Y^prog cell mis-specification into a DA9 neuron. We further used an endogenous *ngn-1* reporter, where GFP has been inserted in-frame at the *ngn-1* endogenous locus. We found *ngn-1* to be expressed only in the WT Y cell, precisely at the time of Td initiation. We thus used it to examine the occurrence of earlier Td initiation in *lin-12^Notch(n676n930)* mutant raised at 15 °C. We never observed any endogenous *ngn-1* expression in the Y cell in newly hatched L1s, either in *lin-12^Notch(n676n930)* mutant or in WT worms (Fig. 8E). Later, at the time of Td initiation, only a small proportion of mutant exhibit a Y cell positive for *ngn-1*, confirming that most animals have no Y cell (Fig. 8E). Furthermore, *ngn-1* timing of expression in *lin-12^Notch(n676n930)* mutant occurs just before Td initiation as in WT, in the few animals where a Y cell is made (Fig. 8E). Next, we investigated whether the expression of PDA markers starts earlier using the canonical early PDA marker, *cog-1*[28,31–33]. In newly hatched L1 *lin-12^Notch(n676n930)* mutant grown at 15 °C, we never observed a Y cell expressing *cog-1* (Supplementary Table 1). Finally, since this putative "precocious PDA" had been observed only in the specific *lin-12^Notch(n676n930)* allele, we

examined other *lin-12^Notch gf* arrays or alleles using several Y and PDA markers (*egl-5, hlh-16* and *cog-1*). We further confirmed that Notch *gf* activity does not influence the temporality of Y-to-PDA Td (Supplementary Table 1). Thus, in early L1s, the Y cell never exhibited an axon or expressed a PDA reporter in any of the genetic backgrounds analysed.

We therefore wondered if the precocious neuron observed by Rashid et al. in *lin-12^Notch(n676n930)* mutant could instead represent mis-specification of the Y^prog cell into a DA9 neuron. To confirm this hypothesis, we further looked at the *ngn-1 nsIs913* transgene used in Rashid et al.[40] whilst also examining a DA9 marker (*mig-13*). In newly hatched *lin-12^Notch(n676n930)* L1 worms, we observed the presence of an *ngn-1* positive cell with an axon as described by Rashid et al. However, this cell consistently co-expressed the DA9 cell marker, both at 15 °C and 25 °C (Fig. 8F–H). In WT worms, we found that the expression profile and dynamics of *nsIs913* is more promiscuous than the endogenously tagged *ngn-1(dev137)* allele. In particular, the endogenous DA9 cell expressed the *ngn-1 nsIs913* reporter at low levels, explaining why the supernumerary DA9 (coming from Y^prog cell conversion) also expresses it in *lin-12^Notch(n676n930)* (Fig. 8F–H). Confirming these findings, two DA9 neurons were observed at a similar proportion later in the development (Fig. 8F) when Td initiates, as in newly hatched L1 mutants, and none of the two DA9 neurons expressed the *ngn-1* reporter anymore (0/55 animals). We observed a very small number of early L1s (3/92) presenting a cell with an axon at the characteristic Y position, that was both *ngn-1* positive and *mig-13* negative ("ngn-1+ neuronal" in Fig. 8F), but at the early L2 stage, this cell was not present anymore. We postulate that these rare events represent failed Y^prog to DA9 conversions. As (i) we and Rashid et al. observed similar proportions of defective Y cells in *lin-12^Notch(n676n930)* mutant at 15 °C (~80%, Fig. 8B, C, Rashid et al.[40]); (ii) we found that the large majority of *ngn-1* positive cells with an axon are also positive for the DA9 marker; (iii) these cells never express a PDA marker; and iv) these cells are not labelled with other Y cells markers, we concluded that the purported "precocious PDA" observed by Rashid et al. is a Y^prog cell converted to a DA9 cell. Overall, these results show that the level of Notch signalling does not regulate the timing of Y-to-PDA Td, but rather the cell-fate decision between a DA9 neuron and a Y rectal cell during embryogenesis, in parallel to its role in Td.

Altogether, our data suggest a model where the level of Notch signalling impacts the number of Y cells formed, and its timing affects the ability of the Y cell to eventually undergo Td. In the absence of the Notch signalling, no rectal Y cell is made (and hence 0 PDA neuron is made). WT levels of Notch signalling have two consequences in Y^prog; it promotes the expression of one set of genes that are necessary to specify Y^prog as a rectal cell; and it leads to activation of another subset of genes, including at least *sem-4* and *hlh-16*, which are not involved in

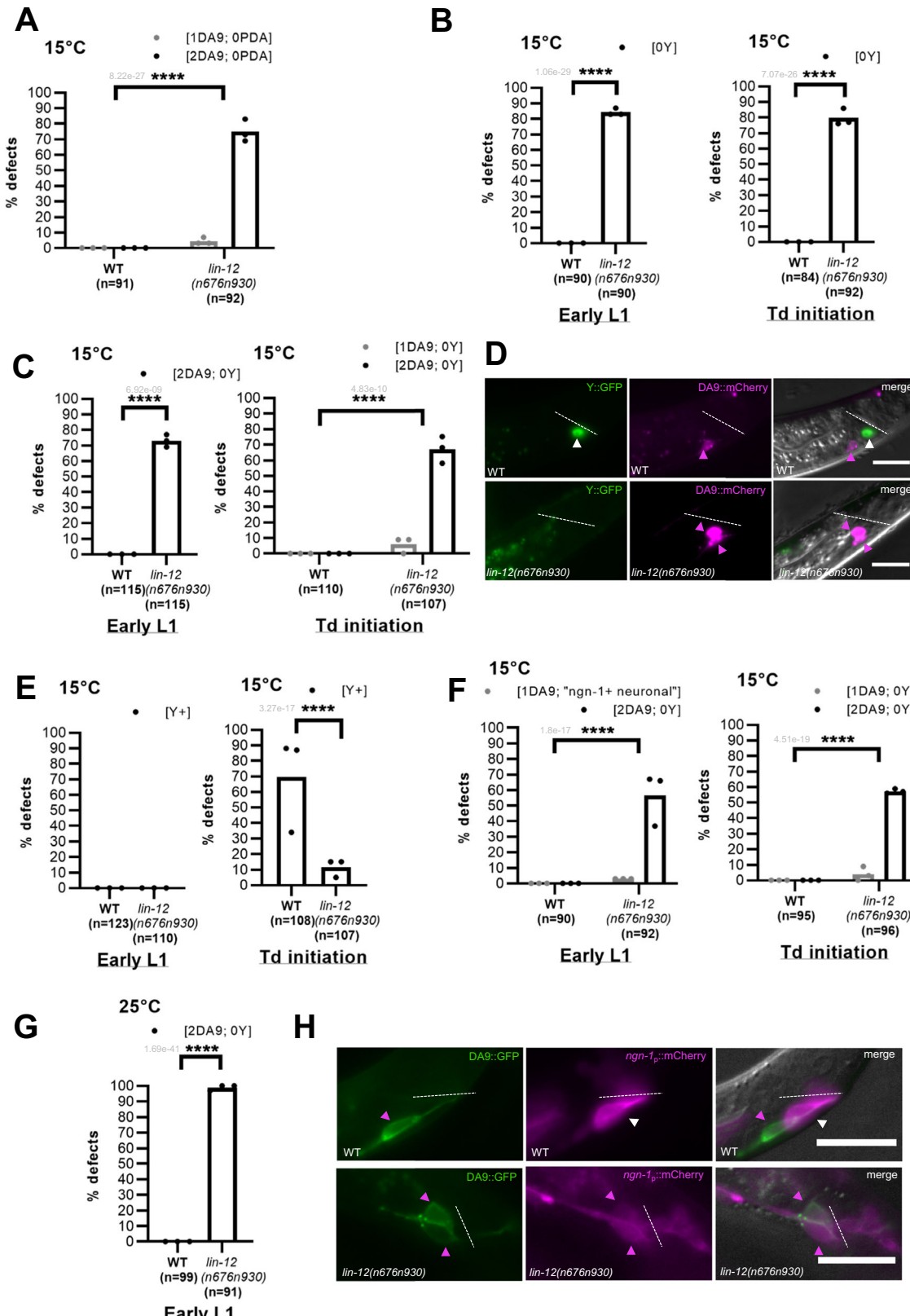

making a Y rectal cell at all, but are necessary to endow it with the competence to undergo Td later on. High levels of the Notch activity at the time of Y^prog birth result in the formation of 2 Y cells that are both competent and later transdifferentiate into 2 PDA neurons. Finally, high levels of the Notch activity closer to Td initiation result in an absence of PDA formation and the Y cell remains rectal (Fig. 9).

## Discussion

This study demonstrates a role of Notch signalling in promoting the competence to transdifferentiate in a naturally occuring Td event. Our findings extend a role for Notch in priming cells for Td, beyond induced settings and across species. Indeed, Notch activity has been previously reported to promote experimentally induced

**Fig. 8 | Notch level regulates the cell-fate decision between DA9 and Y/ PDA fates. A** Quantification (in %) of PDA and DA9 assessed with respectively *cog-1* and *itr-1* reporters in *lin-12*[Notch]*(n676n930)* at 15 °C. **B** Quantification (in %) of Y cell in *lin-12*[Notch]*(n676n930)* at 15 °C assessed with *egl-5* reporter at early L1 stage (left) and early L2 stage (right). [1Y], WT Y cell (without any axon). **C** Quantification (in %) of Y and DA9 cell assessed with respectively *hlh-16* and *mig-13* reporters in *lin-12*[Notch]*(n676n930)* at 15 °C at early L1 stage (left) and early L2 stage (right). **D** Representative picture of [1Y; 1DA9] WT phenotype and [0Y; 2DA9] observed in *lin-12*[Notch]*(n676n930)* L1 animals at 15 °C, assessed with respectively *hlh-16* (Y::GFP) and *mig-13* (DA9::mCherry) reporters. **E** Quantification (in %) of Y cell expressing endogenously tagged *ngn-1(dev137)* allele in *lin-12*[Notch]*(n676n930)* at 15 °C at early L1 stage (left) and early L2 stage (right). Quantification (in %) of Y cell and DA9 neuron in *lin-12*[Notch]*(n676n930)*, assessed with respectively *ngn-1* reporter from

Rashid et al. 2022 (*nsIs913*) and *mig-13* reporters at 15 °C (**F**) at early L1 stage (left) and early L2 stage (right), and at 25 °C (**G**). Except for 3/92 [1DA9; ngn-1+ "neuronal"] animals (**F**), all the cells expressing *nsIs913* always also expressed the DA9 marker. **H** Picture of [1Y; 1DA9] phenotype in WT (top) and [0Y; 2DA9] observed in *lin-12*[Notch]*(n676n930)* (bottom) early L1 at 15 °C assessed with respectively *ngn-1* reporter (*nsIs913*) from Rashid et al.[40] and *mig-13* reporters. **A–C, E–G** *n* total number of animals scored. %, penetrance of the mutant phenotype shown. Data represent the mean of replicates (dots: mean of each replicate). Two-tailed P value, indicated in grey, is calculated using a Chi² test. ****P < 0.0001, ***P < 0.001, **P < 0.01, *P < 0.05. **D, H** Dotted line, rectal slit; white arrowhead, Y cell; magenta arrowhead, DA9 neurons. The rectal area is pictured. Anterior is to the left and ventral to the bottom. Scale bar: 10 µM.

reprogramming of germ cells[53], and also to facilitate the ectopic expression of a specific expression programme in the soma, such as muscle or endodermis, after the forced expression of a relevant cell-fate determinant, in *C. elegans*[54]. In addition, Notch signalling promotes Td after injury in vertebrates (e.g., Td of ingrown lymphatic vessels into blood vessels in zebrafish[55], or Td of murine pulmonary neuroendocrine cells[56]).Thus, it may be that in several different cellular contexts, the Notch signalling triggers the expression of a set of genes that facilitates cell reprogramming.

Our data suggest that Notch acts on two distinct sets of genes that promote independently (i) the rectal identity, and (ii) the rectal-to-neuronal Td (Fig. 9). Accordingly, we find that the *sem-4 /Sall* and *hlh-16/Olig* genes lie downstream of the Notch signalling and strictly belong to the second set. Both genes have been suggested to be a *lin-12*[Notch] target during vulval cell specification (*sem-4*[57]) or neurogenesis (*hlh-16*[39]). Even though several consensus RTGGGAA LAG-1 binding sites[57], the *C. elegans* LAG-1/Su(H)/CBF Notch effector protein, can be found in *sem-4* genomic locus, SEM-4 is only visible in the Y cell well after the LIN-12[Notch] receptor has disappeared (Fig. 4E), a result suggesting that *sem-4* is an indirect target of Notch signalling. However, *hlh-16/Olig* might be a direct target of Notch signalling as its expression onset occurs immediately after *lin-12*[Notch] expression (Fig. 2A, B, 3D). It is likely that the Notch signalling also triggers the expression of a number of other unidentified factors, such as earlier competence-related genes, which may in turn positively regulate later initiation-promoting factors.

In addition, *lin-12*[Notch] signalling could have additional roles to set up an intrinsic environment prone to the erasure of the rectal identity. It could for instance antagonise Polycomb repressive complex 2 (PRC2) activity, as suggested for *glp-1*[Notch] in the promotion of germline fate[53]. However, while we have previously found a role for the histone demethylase *jmjd3.1* in ensuring the robustness of the later re-differentiation phase of Td[33], we have not detected a major role for the PRC2 complex in promoting Y rectal identity or antagonising its ability to initiate a Y identity change (AA, S Zuryn & SJ unpublished), suggesting that the importance of the Notch-PRC2 regulatory relationship may vary depending on the process.

While Notch can convert the DA9[prog] into a transdifferentiation-competent Y cell, most cells in the worm were found refractory to Notch activity, including rectal cells that are seemingly identical to the Y cell, in fact so similar at the transcriptomic level that all rectal cells cluster as one[58]. This highlights the importance of the intrinsic cellular context for the efficiency of a given reprogramming cue, or the existence of barriers mechanisms. The cellular context appears to influence Td in induced settings as well, as overexpression of the muscle determinant MyoD is sufficient to induce the myogenic programme in certain cell types such as adipocytes, but not in all[11]. Similarly, a cocktail of three transcription factors can induce the conversion of pancreatic exocrine cells, but not of the neighbouring cells, into insulin-producing exocrine cells[13]. Why does only DA9[prog]

respond to Notch-induced Td? Making a supernumerary PDA implies two requirements: (1) to have the right toolbox to direct re-differentiation to make a PDA neuron. Indeed, the UNC-3 transcription factor is required for terminal differentiation in both DA neurons and PDA[31,59], suggesting that DA9[prog], but not other rectal cells, could be equipped with the right terminal differentiation machinery to make a PDA. And (2) to have a cell that is permissive to interpret the Notch signal resulting in a competent Y cell. Permissivity could stem from a compatible chromatin context, and/or the presence of plasticity factors. Such permissive context to interpret the Notch signal in the DA9[prog] cell may represent an evolutionary vestige, since it has been suggested that DA9[prog] and Y[prog] were multipotent in an ancestral nematode species[17], and may have remained equipped with part of the inner machinery underlying this multipotency, a machinery now necessary to dedifferentiate (Fig. 1A) and initiate Td. Consistently, the DA9[prog] cell appears to endogenously express several Td initiation-promoting factors, maybe an evolutionary vestige, such as *sox-2, ceh-6/Oct-4*[32]−known to promote pluripotency in mammals[60]−and *egl-5/Hox*, which are required to convert a supernumerary Y cell into an extra PDA neuron. These factors may thus be part of DA9[prog] permissive context. Interestingly, our data reveal that Notch activity can have a dual effect on the destiny of the Y cell (Fig. 9). Notch signalling has been reported to exert either positive e.g.[55,56,61–63] or negative e.g.[64–67] effects on cellular plasticity in different systems. Indeed, Notch has been reported as essential in various lineages for both self-renewal of progenitor cells and differentiation of their descendants[68–71]. However, here we demonstrated that depending on timing, ectopic Notch signalling mediates opposite effects on a single−reprogramming−event, either promoting Td (when endogenous or ectopic Notch is activated during early-embryogenesis) or blocking it (when ectopic Notch is activated during the Td initiation). Such dichotomic action on cellular identity dictates that the timing and duration of LIN-12[Notch] activity should be tightly controlled for proper Td. Our data suggest that this control is at the *lin-12*[Notch] transcriptional level.

How can Notch signalling produce two radically different outcomes? Interestingly, Naylor et al.[6] reporting on the role of Notch in zebrafish kidney development noted a similar dual effect of Notch overtime: while endogenous Notch promoted the Td of renal epithelial cells into Corpuscles of Stannius gland (CS), overexpressing NICD widely and early diminished the size of the CS. Thus, the ability of Notch signalling to produce radically distinct outcomes depending on when Notch activity can be seen during a reprogramming event in vivo might be a conserved theme. It is possible that in both cases, endogenous Notch signalling results in the activation of 2 sets of genes with antagonistic activity that trigger a tug-of-war between differentiation and reprogramming. A different gene balance at different time points of the cell's life may favour one set over the other. Such gene balance could solely rely on dynamic temporal expression [e.g., a−early Notch targets expression specifying Y rectal identity and Y Td competence; and, b−downregulation of the Y-identity signal plus response to

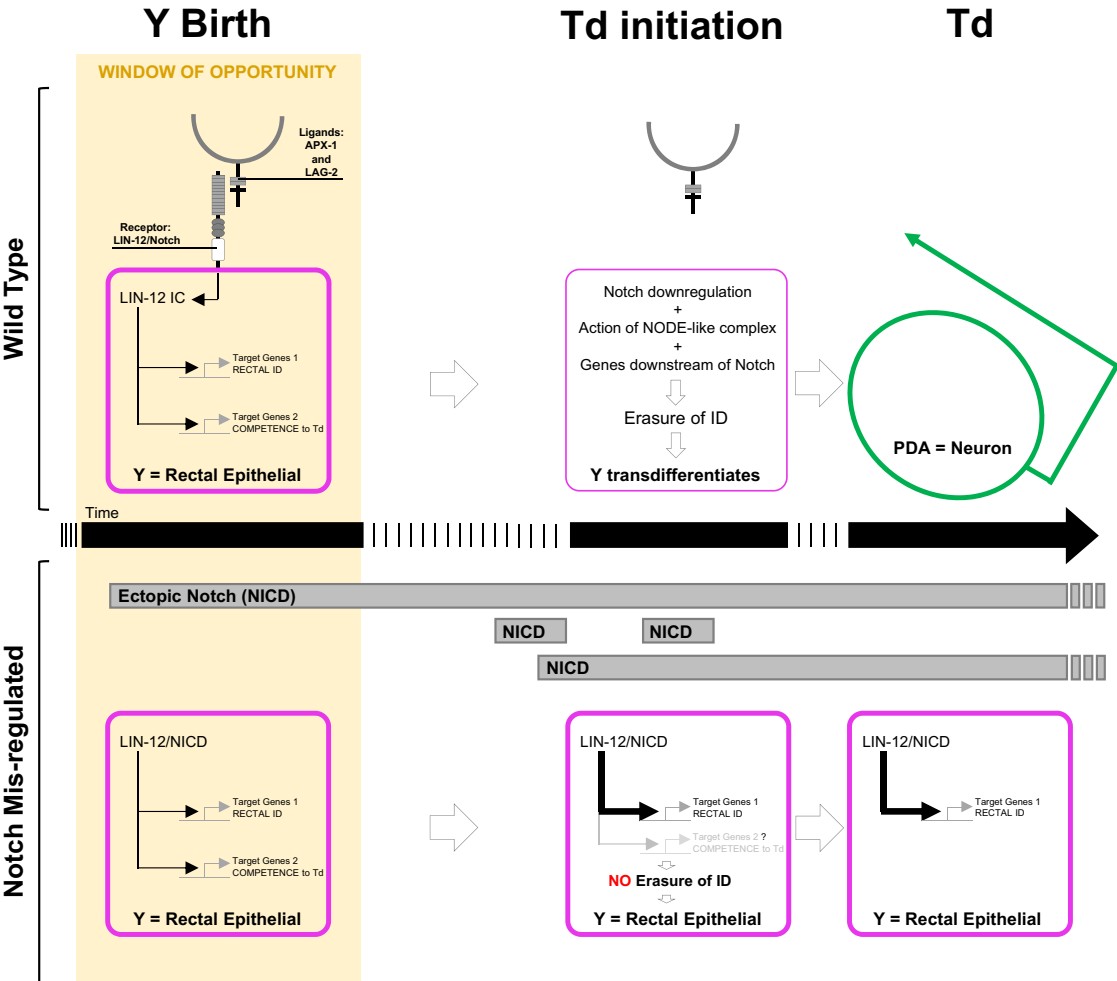

**Fig. 9 | Model for the impact of Notch activity on natural Y-to-PDA.** In the WT situation, a pulse of Notch triggered by two redundant ligands (*apx-1* and *lag-2*) around Y birth will activate two potential sets of genes, endowing the Y cell with its rectal epithelial identity as well as its competence to transdifferentiate. The action of these 2 sets can be distinguished, as at least two genes downstream of Notch are crucial for the initiation of Td while not impacting on Y identity. This time interval of Notch signal represents the window of opportunity during which the Notch pathway can promote Td. In WT animals, Notch signal is then down-regulated at the transcriptional level to allow the Y cell to initiate Td, the first step being the erasure of its initial identity. When a later Notch signal is provided to the Y cell, it does not reset a timer and delay Td, but results in a block of Td by over-imposing in Y a rectal epithelial fate. Together, it shows how one signalling pathway, Notch, can exert opposite effects if activated at different steps, on Y-to-PDA Td, a step-wise process that occurs in the absence of cell division. It further shows that both the extrinsic environment, as well as the intrinsic cellular context, combine to allow one cell to switch identity.

Td-promoting]. However, our data highlight the existence of a window of opportunity for Td to occur, as even a pulse of late ectopic Notch signalling results in a definitive block, rather than in a reset of the timing of the process. Thus, the Y[prog] intrinsic context may evolve over time, from initially permissive to the extrinsic Notch signal endowing the Y cell with the competence to transdifferentiate, to not anymore. Indeed, once rectal, Y appears to only allow implementation of Notch-induced rectal identity instructions, making it so stable that it becomes a barrier to reprogramming.

Does Notch signalling contribute more directly to the initiation of Y Td much later in development? We have ruled out that the levels of this transient extrinsic signal are involved in setting the time of Td initiation. The endogenous Notch pulse in the Y cell could either simply empower the Y cell with the ability to change its identity but have no bearing on the actual initiation of Td; act jointly with a later, unidentified, signal to trigger initiation of Td; or could trigger cell-autonomously a long cascade of events that ultimately result in a dedifferentiation. Further work will be needed to determine what lies downstream of Notch activation and discriminate between these hypotheses.

While biological processes, such as cell-fate acquisition, are often seen as linear and progressive events, our findings once more highlight the importance of the exquisite and highly dynamic regulation of signalling over time. Importantly, our data emphasise how a prolonged (or *out-of-time*) Notch signal could lead to the opposite phenotype to WT, potentially biasing conclusions on Notch role obtained by inducing prolonged ectopic Notch signalling. These findings have significant implication for the experimental design of studies aimed at deciphering the role of Notch signalling in biological processes. Therefore, the temporality and dynamics of the process studied must be carefully assessed and taken into account.

## Methods

### *C. elegans* maintenance, strains and alleles

Standard methods as described in Brenner 1974[72] were used for worm handling, maintenance and genetic analysis. Experiments were mostly performed on hermaphrodites at 25 °C unless otherwise indicated. The wild-type parent for most strains used in this study is the *C. elegans* var. Bristol strain N2. See Supplementary table 2 for a list of all strains used in this study.

## Microscopic observations and cell identification

DIC and epifluorescence observations were performed using a Zeiss Z1 imager microscope using a Hamamatsu Orca-ER camera C4742 or a Leica DM6 B microscope with LAS X software and the Hamamatsu Digital Camera C11440. Worms were mounted on a 2% agarose pad and anaesthetised with 50 mM of sodium azide. For all images, anterior is to the left and dorsal up. Single focal planes, or maximum projections when the cells of interest did not coexist on the same focal plane, were used. To assess cell identity, the following criteria were used: for PDA identity (WT PDA or "2 PDA" phenotypes), DIC speckled nuclear morphology (note that PDA identification only by DIC is difficult), WT final position, PDA marker expression [cog-1p::GFP], presence of an axon projecting posteriorly, then antero-dorsally, as found in WT L3 and older worms. DA9 identity: WT final position, DA9 marker expression [itr-1p::mCherry or mig-13p::mCherry], presence of an axon projecting posteriorly, then antero-dorsally, as found in WT L4 and older worms. ABplpppaaaa (DA9$^{prog}$), ABprpppaaaa (Y$^{prog}$) and Y cells: DIC "fried egg" nuclear morphology, WT position, marker expression. Of note, another nearby embryonic cell very transiently expresses translational (but not transcriptional) hlh-16 reporter between -1.7-fold and 3-fold embryonic stages. Our scorings in lin-12$^{Notch}$(gf) backgrounds were performed before this endogenous expression. Expression of hlh-16 is maintained in the Y cell until the Td initiation (L2 stage), in WT animals. Fluorescent protein expression was used to follow over-expression of NICD, and mosaic expression associated with non-integrated Ex arrays was taken advantage of to analyse cellular focus of action. We have assessed Notch ability to confer the competence to transdifferentiate using PDA markers as final output. While we did not observe a wide plasticity across the worm's cells, we note that other reprogramming events that would lead to the creation of cells other than a PDA neuron, would go unnoticed. Further studies using a battery of terminal neuronal identities, and other non-neuronal identities, will be required to address this point.

## RNAi experiments

RNAi experiments were performed as previously described[73]. Double-stranded RNA (dsRNA) was injected in the relevant genetic background combined with the RNAi hypersensitive mutation rrf-3(pk1426). DsRNA was obtained from in vitro transcription of PCR fragment corresponding to cDNA or genomic matrices: egl-27, sem-4, egl-5 and hlh-16 matrices from Ahringer-MRC feeding RNA interference library[74]: all clones used from this library were first sequence-verified; ceh-6 target 1 and sox-2 target 2 matrices from Kagias et al.[32]; sel-12: PCR amplification from genomic DNA using the following primers:

F-T7promoter – 5′ taatacgactcactatagggATGCCTTCCACAAGGA-GACAAC 3′ and R-T7promoter−5′ taatacgactcactatagggGAGATCGCT CAAGATATAATCGAAAAG 3′. In vitro transcription was performed using the PCR products as templates with T7 RNA polymerase using the mMESSAGE mMACHINE™ T7 Transcription Kit (Invitrogen™). Single-stranded RNA was annealed to form dsRNA by gradually lowering the temperature of the sample from 65 °C. In vitro transcripts were purified on RNeasy columns (Qiagen) and injected into the gonads and pseudocoelom of young adult worms. F1 progeny derived from these adults were scored for the presence of the Y, PDA and DA9 cells.

## Heat-shock experiments

All the strains used for the heat shock experiments were maintained at 20 °C. Fast induction of fluorescent construct was achieved by raising the temperature as described before[31] and briefly below.

**Induction of NICDGFP expression around Y birth.** To obtain sufficient numbers of animals to score, egg pulses were performed with minimum 100 gravid transgenic hermaphrodites at 20 °C for 1 h on plate with food. Mothers were removed and embryos were left on plate for 15 min, when most embryos were between 165 min and 225 min of development. Plates with embryos were sealed and immerged in a water bath at 34 °C for 30 min. The embryos were immediately cooled down in a 20 °C water bath and left to grow at 20 °C until the L4 stage, when they were scored. Both transgenic and non-transgenic siblings were handled and scored in parallel.

**Induction of NICDGFP expression at mid-embryogenesis.** Egg pulses were performed with 100 gravid transgenic hermaphrodites at 20 °C for 1 h on plate with food. Mothers were removed and embryos were left on plate for 3 h. Plates were further sealed and immerged in a water bath at 34 °C for 30 min. The embryos were immediately cooled down in a 20 °C water bath and left to grow at 20 °C until the L4 stage, when they were scored. Both transgenic and non-transgenic siblings were handled and scored in parallel.

**Induction of NICDGFP expression before transdifferentiation initiation.** Eggs obtained from gravid mothers were allowed to hatch overnight in M9 without food. Synchronised L1 were then put on plate with OP50 for 8 h before heat shock at 34 °C for 30 min. Worms were immediately cooled down in a 20 °C water bath and left to grow at 20 °C until the L4 stage, when they were scored. Both transgenic and non-transgenic siblings were handled and scored in parallel.

## Temperature-shift experiments

**Temperature-shift experiments of glp-1(ar202ts) and glp-1(e2141ts).** Egg pulses were performed on pre-warmed plates at 25 °C for 1 h with gravid hermaphrodites grown at 15 °C. Mothers were removed and embryos were left at 25 °C (the restrictive temperature) until the adult stage, to score for PDA phenotype and to monitor gonads sterility and morphology.

**Temperature-shift experiments of apx-1(zu347ts) and lag-2(q420ts).** Non-sensitised and sensitised backgrounds were treated the same way during temperature-shift experiments. Worm populations grown at 15 °C were bleached. The resulting embryos were put on pre-warmed plates at 26.5 °C without food. After 7 h, all the synchronised hatched L1 larvae were washed away with M9 and discarded, to remove animals that would have been shifted to the restricted temperature too late. OP50 was added subsequently to the plates to allow growth until the L4 stage, when they were scored. Note that the double mutant lag-2(q420ts) apx-1(zu347ts)V could not be built as the two mutant alleles are genetically very close (0.18 cM).

**Temperature-shift experiments of lin-12$^{Notch}$(n676n930).** Non-sensitised and sensitised backgrounds were treated the same way during temperature-shift experiments. Worms were placed at 15 °C or 25 °C tightly controlled incubators two generations before scoring. Gravid mothers were bleached and put in suspension in M9 overnight at 15 °C or 25 °C and place on plates with food the next morning. Worms were directly scored for early L1 scoring, or let at 15 °C 23.5 h post L1 for early L2 scoring, as described in Rashid et al.[39].

## SNP mapping of hlh-16(fp12)

After the cross between Hawaiian strain WT males and N2 strain fp12 hermaphrodites, 20 F2 worms with "0 PDA" has been isolated (recombinants fp12/fp12 homozygotes). Specific SNPs to Hawaiian and N2 strains on chromosome I (positions I-19, I-12, I-6, I5, I14 and I26 cM) have been analysed by PCR/restriction in every 20 recombinants.

## Transgene generation

C. elegans extrachromosomal transgenic strains were created by DNA microinjection in the gonad of young adults[72] of the plasmid of interest together with a co-injection marker and pBSKII$^+$ to a final

concentration of DNA of 200 ng/μL in water. See Supplementary Tables 3 and 4 for a list of all extrachromosomal arrays and integrated arrays, respectively, generated in this study.

*hlh-16(syb683[GFP::linker::hlh-16])* (SunyBiotech) was designed by Christelle Gally.

## Plasmid generation

See Supplementary Table 5 for a list of all oligonucleotides used in this study.

pSJ3177−*MCS::NICDGFP::unc-54 3'UTR*: NICDGFP was amplified by PCR from the plasmid pSJ201 – *NICDGFP in pBSKII⁺* with the following primers:

F-KpnI−5′ AAC GGTACC AGAAAAAATGGTTGTTCTGATGTTAGGA GCATTACC 3′

R-KpnI−5′ TT GGTACC TCAAAAATAATGAGCTGGTTCGGAGTATC G 3′

The obtained PCR product was digested with KpnI and inserted in the multiple cloning site (MCS) 2 of pSJ901 – *MCS1::MCS2::unc-54 3UTR'*.

pSJ3171 - *hsp-16.2::NICDGFP::unc-54 3'UTR: hsp-16.2* was excised from pPD49.78 with BamHI and HindIII, and was inserted in MCS1 of pSJ3177 – *MCS::NICDGFP::unc-54 3'UTR'*.

pSJ6003 - *egl-5(6,2 kb)Δpes10p::NICDGFP::SL2::mCherry::unc-54 3' UTR:* NICDGFP was amplified by PCR from the plasmid pSJ201 with the following primers:

F-KpnI−5′ AAC GGTACC AGAAAAAATGGTTGTTCTGATGTTAGGA GCATTACC 3′

R-KpnI−5′ TT GGTACC TCAAAAATAATGAGCTGGTTCGGAGTATC G 3′

The obtained PCR product was inserted in the MCS of pSJ671 - *egl-5(6,7 kb)Δpes10p::MCS::SL2::mCherry* at a unique KpnI site.

pSJ3173−*col-34p::NICDGFP::unc-54 3'UTR: col-34p* was amplified by PCR from genomic DNA with the following primers:

F-SphI−5′ ACA GCATGC GACATGTAAAGTACATCCGTTACATC 3′

R-XmaI−5′ CCC CCCGGG TGTATGCAGTGGTGGTTTGG 3′

The obtained PCR product was digested by SphI and XmaI, and inserted in the MCS of pSJ3177 – *MCS::NICDGFP::unc-54 3'UTR*.

pSJ3169−*lin-48p::NICDGFP::SL2::mCherry::unc-54 3'UTR: lin-48p* was amplified by PCR from genomic DNA with the following primers:

F-*SphI*−5′ ACAT GCATGC GGATCCAAAAAACCTGCATTTTTTTCAG 3′

R-*XmaI*−5′ CCC CCCGGG CTGAAATTGAGCAGAGCTGAAAATTTTT G 3′

The obtained PCR product was digested with SphI and XmaI, and inserted in the MCS of pSJ3177 – *MCS::NICDGFP::unc-54 3UTR'* resulting in pSJ3169 – *lin-48p::NICDGFP::unc-54 3UTR'*

The SL2::mCherry sequence was amplified by PCR from pJG7-psm-SL2-Mcherry (Gift from Cory Bargmann laboratory) with the following primers:

F-NotI−5′ GCGGCCGC GCTGTCTCATCCTACTTTCACC 3′

R-NotI−5′ GCGGCCGC CCTACTTATACAATTCATCCATGCC 3′

The obtained PCR product was digested by NotI and inserted after NICDGFP in *pSJ3169 – lin-48p::NICDGFP::unc-54 3'UTR*.

pSJ3162−*egl-20p::NICDGFP::SL2::mCherry::unc-54 3'UTR: egl-20p* was amplified by PCR from genomic DNA with the following primers:

F-SphI−5′ AAA GCATGC GAAGTCATCCTACTAACTAACAATATGAC GC 3′

R-XmaI−5′ AAA CCCGGG TATTTCTGAAATTGAGATGTTTTAGAAT TTC 3′

The obtained PCR product was digested with SphI and XmaI, and inserted in the MCS of pSJ3177 – *MCS::NICDGFP::unc-54 3'UTR* resulting in pSJ3164 – *egl-20p::NICDGFP::unc-54 3'UTR*

SL2::mCherry sequenced as been inserted as previous i*n pSJ3164 – egl-20p::NICDGFP::unc-54 3'UTR*.

pSJ3103−*egl-5(6,2 kb)Δpes10p::NICD::SL2::mCherry::unc-54 3'UTR*: The GFP of the NICDGFP sequence was deleted by inverse PCR on

pSJ6003 - *egl-5(6,2 kb)Δpes10p::NICDGFP::SL2::mCherry::unc-54 3UTR'* with the following primers:

F-5′ GACTCAACTCATCTGACACCTCC 3′

R-5′ GGGTCGAGTTACTTTTCTTGAAGG 3′

pSJ3215−*col-34p::lin-12cDNA::unc-54 3'UTR and* pSJ3217 - *col-34p::lin-12cDNA::SL2::mCherry::unc-54 3'UTR: lin-12* cDNA was amplified from pLM2.4 – *lin-12cDNA in pBSKII⁺* (a gift from the Greenwald laboratory) with the following primers:

F-KpnI−5′ AAAAGGTACCATGCGGATCCCTACGATTTG 3′

R-KpnI−5′ TTTTGGTACCTCAAAAATAATGAGCTGGTTCGG 3′

NICDGFP was excised by KpnI from pSJ3173 – *col-34p::NICDGF-P::unc-54 3UTR'* and pSJ3148 - *col-34p::NICDGFP::SL2::mCherry::unc-54 3UTR'* and replaced by the obtained PCR product of *lin-12* cDNA digested by KpnI.

pSJ3218−*col-34p::lin-12cDNA(n137)::unc-54 3'UTR:* TCT to TTT point mutation was generated by site-directed mutagenesis on pSJ3215 - *col-34p::lin-12cDNA::unc-54 3UTR'* with the following primers:

F-*n137*−5′ GTGTTGTTGACTCAATA TTT GCAAGGCTTGC 3′

R-*n137*−5′ GCAAGCCTTGC AAA TATTGAGTCAACAACAC 3′

pSJ3222 – *col-34p::lin-12cDNA(n941)::*unc-54 3'UTR: TGG to T*AG* Premature stop codon was generated by site-directed mutagenesis on pSJ3215 - *col-34p::lin-12cDNA::unc-54 3UTR'* with the following primers:

F-*n941*−5′ GGATTCGGTGGGAAA TAG TGTGACGAGCCATTG 3′

R-*n941*−5′ CAATGGCTCGTCACA CTA TTTCCCACCGAATCC 3′

pSJ3223−*col-34p::lin-12cDNA(ΔANK)::unc-54 3'UTR*: Deletion of the seven ankyrin repeats contained in the intracellular part of the *lin-12* receptor was performed by inverse PCR according to the recommendations of Rhett Kovall with the following primers:

F-5′ CCAGAACGAGAATATTCAATGGATC 3′

R-5′ AGGTTCAGGTTCAGTTGGAATTTG 3′

pSJ3212−*lin-12p::NICDGFP::lin-12UTR:* A megaprimer containing NICDGFP was amplified from pSJ3177 with the following primers:

F-5′ CTCAACAGACTTTGCTCAATTTCAAAAAATGGTTGTTCTGA TGTTAGGAGCATTAC 3′

R-5′ GGAATTTAAATAATAAATGACGATTGTTCAGAAGATGTACCG AGCTCGGATCCACTAGTAAC 3′

The obtained megaprimer was used to replace the GFP from the plasmid pSJ3143 - *lin-12p::GFPpest-lin-12 1st intron::U54UTR* by Overlap extension PCR cloning.

pSJ3240−*lin-12p(ΔR1)::NICDGFP::lin-12 UTR:* The first conservation region of the *lin-12* promoter was deleted by inverse PCR on pSJ3212−*lin-12p::NICDGFP::lin-12UTR* using the following primers:

F-5′ ACAGTAACAGACACCTGTGCTCC 3′

R-5′ TATTGTTAATAAATGAGTGTAACATTTAAG 3′

pSJ3242−*lin-12p(ΔR2)::NICDGFPmut::lin-12 UTR:* The second conservation region of the *lin-12* promoter was deleted by inverse PCR on pSJ3212 – *lin-12p::NICDGFP::lin-12UTR* using the following primers:

F-5′ ATTAATGATAATGCAAAAGCTACCAGG 3′

R-5′ TGTGTCAGTTTTTAGAGTTTTATTTCTG 3′

pSJ821−*hlh-16p::mcherry::hlh-16::hlh-16 3'UTR:* Translational *hlh-16* reporter from Bertrand et al. [38] (a gift of Vincent Bertrand/Oliver Hobert) in which GFP was swapped for mCherry.

pSJ823−*col-34p::mcherry::hlh-16::hlh-16 3'UTR: hlh-16* promoter from pSJ821 was replaced by *col-34* promoter.

pSJ6334−*ceh-6p::GFP [Transcriptional reporter]:* A 5.6 kb *ceh-6* promoter fragment was cloned in front of GFP using the following primers:

F-5′ ATAAGAATgcggccgcCGTGTTGCTTTAGCACTTCTCCATCCC TTC

R-5′ ATAGTTTAgcggccgcCAGTTGGGAAGTCCAGGAGCAACGGG GTG

A 3.6 kb *ceh-6* 3'UTR fragment was cloned after GFP sop codon using primers:

F-5′TTTTTTGTGATGCGTATTGATGTAGC

R-5′ GTCGACACAGAAACTACGCAAAATC

_pSJ1007 - egl-5p(6 kb)::2xNLS::mCherry::unc-54 3'UTR_: _2NLS_ was added to pSJ671 with the following primers:

F-5′   CGAGCTCAGAAAAAATGACAGCACCGAAAAAAAAGCGAAA AGTTCCAGCTGAGAAGATGACCGCTCCAAAGAAGAAACGCAAAGTA 3′
R-5′      CCGGTACTTTGCGTTTCTTCTTTGGAGCGGTCATCTTCT-CAGCTGGAACTTTTCGCTTTTTTTTCGGTGCTGTCATTTTTTCT-GAGCTCGGTAC 3′

_pSJ834−egl-5(1,3 kb)delta pes10::mkate::unc-54 3'UTR:_ mkate was added to pSJ6273 using the following primers:

F-5′ TAGAGGATCCccgGGGGATTGGCC
R-5′ GATATTATACATATTTCATAAAGCCAACC

_pSJ503−exp-1p::mcherry:_ an ApaI-BamHI fragment from PD95.75 and containing _mCherry::unc-54 3UTR'_ was cloned into the GM4 vector (_exp-1_ promotor).

## Statistical analysis

Statistic test was always performed between wild type and mutant. The stars summarise the statistical significance as calculated through the Chi$^2$ test. As the Chi$^2$ test does not take into account the replicates, no SD or SEM are showed on the graphical representations. However, the data for each experiment are represented by dots on the bar graphs. *$p < 0.05$; **$p < 0.01$; ***$p < 0.001$; ****$p < 0.0001$ and ns, not significant, for all the tests.

## Reporting summary

Further information on research design is available in the Nature Portfolio Reporting Summary linked to this article.

## Data availability

The published article includes all datasets generated or analysed during this study. Supplementary figures are available in the Supplementary Information file. Further information and requests for resources and reagents should be directed to and will be fulfilled by Sophie Jarriault (sophie@igbmc.fr). Reagents generated in this study will be made available on request, but we may require a payment and/or a completed Materials Transfer Agreement if there is potential for commercial application. Source data are provided with this paper.

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

## Acknowledgements

We thank James Priess, Geraldine Maro & Keng Shen, Xiao Liu, Shai Shaham, Stuart Kim and Vincent Bertrand for reagents, Christelle Gally for designing *hlh-16::GFP* CRISPR insertion, Steven Zuryn & Thomas Kleiber for help with experiments and Richard Poole, Peter Meister, Christelle Gally, Thomas Le Gal, Eria Becker and Deborah Warrington for comments on the manuscript. This work was supported by a Ministère de l'Enseignement Supérieur et de la Recherche (to T.D. and J.C.) and an Association pour la Recherche sur le Cancer predoctoral fellowships (to T.D.); and grants from the Fondation pour la Recherche Médicale (DRC20091217181), the Agence Nationale pour la Recherche (CELLS-witch; ANR-13-BSV2-0005), the Association Française contre les Myopathies (#15352) and the Association pour la Recherche sur le Cancer (#SFI20121205880), to S.J. S.J. is a research director of the CNRS.

## Author contributions

T.D.: conceptualisation, methodology, investigation, formal analysis, original draft, writing - review & editing, visualisation; J.C.: conceptualisation, methodology, investigation, validation, formal analysis, original draft and revised draft, writing - review & editing, visualisation; MCM: methodology, investigation, formal analysis; A.A.: methodology, investigation, formal analysis; D.I.: methodology, investigation, formal analysis; S.J.: conceptualisation, investigation, formal analysis, resources, writing - original draft and revised draft, writing - review & editing, supervision, project administration, funding acquisition.

## Competing interests

The authors declare no competing interests.
