## [Peer Review File · Nature Communications]

REVIEWER COMMENTS

Reviewer #1 (Remarks to the Author):

This manuscript offers several notable conceptual and molecular advances in our understanding of the Y-to-PDA transdifferentiation (Td) model and their relationship to Notch signaling dynamics. Overall, the data support a model in which the temporal regulation of Notch activity and that of specific suites of downstream factors enable a sequential series of events such that (1) a relatively early pulse of Notch signaling activity in embryogenesis both specifies the Y fate (versus DA9) and triggers the expression of genes required later for Td and (2) the timely down-regulation of Notch signaling relaxes the Y fate and allows Td to occur. In particular, the series of experiments expressing Notch NICD from promoters that are (*lin-12p*) or are not (*egl-5p* or *col-34p*) downregulated are compelling. In addition, a new allele of *hlf-16* is identified, the relevant ligands for the Notch receptor are identified, and the mechanism for downregulation of Notch is demonstrated to be transcriptional rather than post-transcriptional or due to changes in ligand availability.

Finally, the authors present copious evidence to counter a previous study suggesting that the timing of Td is affected by Notch signaling levels, providing additional results demonstrating that the *n676n930* allele is not acting in a *gf* manner at 15° as it does in other phenotypic contexts. Rather it acts as a *lf*. Here they argue persuasively, using a battery of markers, that a 2 PDA phenotype is never observed, but rather a 0 Y (and PDA), 2DA9 is observed as would be expected for a loss or reduction of *lin-12* activity, and indicating with markers that the previously reported precocious PDA was actually DA9.

The data are robust and are presented clearly. While direct demonstration of temporally regulated degradation of *egl-5p* NICD activity (e.g. by degron or by RNAi) and observation of the expected restoration of the 2 PDA phenotype is not presented, sufficient convincing independent approaches are provided to support the model.

Nevertheless, in several places the text is needlessly confusing and requires revision.

1. Title is confusing – particularly “positive selection” and “defined window of opportunity”—and should be changed to reflect the main findings in a more direct way.
2. Throughout, the authors can be more specific when referring to the previous literature. For example, several places “another cell” or “another player” or “another gene” could just state which cell, player and which gene(s).
3. Heatshock promoter expression is not evenly ubiquitous. This must be stated more carefully.

4. The term “Notch signal” should be avoided since it can be misunderstood as a reference to the ligand. “Notch signaling” is better.
5. It is important to mention the incomplete penetrance of the phenotypes in the text as well as in the figures; alert the reader to what they should focus on (e.g., the appearance of any 2 PDA vs all 1 PDA).
6. The terms describing gene expression experiments need to be much clearer regarding temporal versus spatial (or both) alterations in gene expression. Throughout the manuscript, the terms “ectopic, exogenous, extrinsic, intrinsic” are often confusing and need to be replaced or better qualified. For example, in cases of heterologous gene expression, it may be best to state the promoter and its presumed temporal and spatial expression pattern.
7. There is a lot of interesting information about hlh-16 in the section starting line 148; consider a separate section?
8. The term dichotomic is very confusing. Even “paradoxical”, which is better, could be recast as “seemingly paradoxical”. However, both are based on the history of the field. Once the model is presented, it is not so paradoxical, but rather all about temporal regulation of lin-12 targets (direct or indirect, maintained or not). The story may be less confusing if it were possible to very clearly present the expectation, the deviation from expectation, the hypothesis, and then the data that support it. That is, based on lin-12p experiments, and the similar onset of expression of egl-5p in all rectal cells including DA9 and Y, you would have expected egl-5p::NICD in Y to have at least 1 PDA phenotype, if not 2 PDA but saw 0 PDA. You interpret 0 PDA as failure of the specified Y to Td into PDA. Given that egl-5p expresses over a longer timeframe than lin-12p, you then hypothesized that the perdurance of NICD could block Td and tested this directly with col-34, etc. Here a “window” concept would be replaced with dynamic temporal expression concept (as is your model), with two sets of targets: (1) early Notch activity and targets specify Y identity (versus DA9) and competence for future Td (2) the necessity to down-regulate the Y-identity signal and allow Td to occur in response to Td-promoting targets triggered by early N. While I appreciate the attempt to mostly provide the data before providing the model, the situation here is sufficiently complex that it would be worth considering the benefit to the reader to suggest the model and then provide evidence. I would also refer the reader to the discussion in the text of the results as the points are further well-developed in the discussion.
9. The argument for a “permissive context” (line 224 section) was not clear.
10. I found the introduction of DA9prog and Yprog terms to be confusing. Maybe “future DA9”?
11. Related to point above, reword line 553-558, dichotomic, etc. and later in the next paragraph. It is not that Notch activity is blocking Td in the natural context; this only occurs in the experimental situation. Rather, it is promoting Td by activating hlh-16 and sem-4 earlier.
12. The figure presented in SI Fig 3A was extremely helpful to understand the temporal dynamics of the system and the experiments and should be considered as a main figure panel in some form.
13. Additional specific points:
 - a. line 51, will transdifferentiate;

- b. line 52, certain cell types;
- c. line 74, replace “make” with “specify”?
- d. line 91, not really a paradoxical action on the Y fate – just paradoxical with respect to the expected phenotypes
- e. line 94, delete “on”
- f. line 100-101, given points above, reconsider title – ectopic, induce, extra cell
- g. line 113, specify other mutant backgrounds
- h. line 126-127, reconsider use of exogenously, ubiquitously
- i. line 131, not supernumerary Td, but supernumerary PDA – here and later, supernumerary refers to the number of something, not a process
- j. line 159, reconsider vague term “assorted”
- k. line 224, reconsider section title
- l. line 243 and section, reconsider “dichotomic”
- m. line 296, reconsider title to be more clear (Notch activity, “it”)
- n. line 321, reconsider title
- o. line 384, dichotomic
- p. line 524, “another cell”

Reviewer #2 (Remarks to the Author):

In the provided manuscript, the authors report on how Notch signaling affects the transdifferentiation of rectal epithelial cells in the nematode *C. elegans*.

Natural Transdifferentiation phenomena such as the direct conversion of the rectal epithelial cells Y to the motoneuron PDA in *C. elegans* offer an excellent research system to elucidate the molecular dynamics of cell fate conversion processes. Moreover, this particular natural transdifferentiation instance allows investigations at the single-cell level in vivo with high accuracy.

The group of S. Jarriault has been central in dissecting the above-mentioned cell fate conversion process and pioneered a large number of critical findings. In this study, they defined the temporality of Notch signaling, a highly conserved signaling pathway relevant to cell fate specification and many other developmental processes. Interestingly, in previous studies, NOTCH signaling has been

implicated in cellular reprogramming as either an inhibitory or promoting pathway depending on the context of reprogrammed cells and their niche, and species. Yet, this study by Daniele et al. reveals in a very convincing manner that NOTCH has a dichotomic effect on the same cellular reprogramming instance, depending on timing. This outcome is highly relevant for the reprogramming research field and prompts for more careful assessment of cellular signaling pathways that affect reprogramming efficiencies in general.

Moreover, the meticulous work by Daniele et al. in this study also clarifies an observation made by a previous study from Rashid et al. concerning the timing of PDA neuron fate induction in the lin-12 Notch(n676n930) mutants background that supposedly has a gain-of-function effect at 15°C. Rashid et al., reported that gain of Notch function leads to a precocious Y to PDA conversion. Yet, Daniele et al. demonstrate in a very convincing manner, that the Y cell started expression DA9 cell marker. Examining the DA9 neuron cell marker mig-1351 they saw that lin-12 Notch(n676n930) worms raised at the same temperature as in the Rashid et al. study show two DA9 neurons but no Y cell. Hence, they conclude that the progenitor Y cell is mis-specified to a DA9 neuron rather than becoming a PDA neuron precociously.

The study has been performed using an impressively large number of different worm strains with different genetic backgrounds and reporter transgenes, highlighting the effort that has been undertaken to elucidate gene expression and cell fate induction dynamics and temporality with high accuracy.

Overall, this study is of high value for a variety of research fields dealing with development, cellular reprogramming, and Notch signaling. I have only a few comments and questions that should be addressed before publication.

Comments and questions to the authors:

1. Line 113 – 114:

Explain or mention briefly “...other mutant backgrounds do not have the capacity to transdifferentiate into a PDA neuron.”

2. Figure 1:

3-Fold embryo in panel G has very limited capacity to provide any information – near useless as the movement of the animal causes blurring and mixing of signals during the exposure time. The use of a bright GFP-based injection marker combined with the actual GFP-reporter is very unfortunate. A workaround instead of making a new strain could be to fix the animals briefly and / or stain them with anti-GFP Abs.

3. Strain list:

The long list of worm strains used is impressive and at the same time makes it difficult to easily find information about strains and transgenes. E.g. what is the exact nature of the *hlh-16::gfp* reporter?

4. Notch signaling:

A. Is it sufficient to have Notch only during embryo development (depleted starting L1) to allow Y to PDA conversion?

B. Is the NICD always transported to the nucleus – this may be important to know whenever NICD (over)expression does not show an effect.

C. The question before and other matters could be addressed if something like a ‘Notch sensor’ is possible. Meaning, a reporter construct that bears several copies of binding sites for the Notch effector protein. This reporter could be used to monitor whether the examined cells and time points are indeed ‘receiving’ effective Notch signaling (also useful for potential non-cell-autonomous effects).

5. Figure 5:

Panels A,E axis labels are hard to read – too small!

6. Lines 375 -377

Statement about Notch conserved domain appears misplaced in this study.

7. Line 399

The sentence reads difficult – ‘made’ at the end of the sentence, correct?

8. Some parts of the discussion, like at the beginning, are too long or redundant with other sections. Overall, it would help the manuscript to be a bit more concise.

We thank the referees for their praise, and insightful and constructive comments. Please find below our answer point by point and the associated text changes. These can also be visualized in the manuscript text file with the track changes. In particular, the discussion has been re-organised - and shortened - to remove passages that felt redundant. We also have changed the title following one of the referee's comments and so as to fit the Nature Communications length limit.

In addition, we have made the following changes to fit Nature Communications formatting:

- Making sure the formatting of the figures is as recommended by Nature Communications guidelines (font, size, style, etc)
- Changing the set of the figures' colours using an alternative to a red&green scheme
- Adding the experimental points on the graph bars, where each point represents the mean of one replicate
- Shortening the Figures' legends
- We clearly specified that the research was performed on hermaphrodites, in the abstract as well as in the Mat & Meth (see line 31 for instance)

All the ORCIDS should have been entered as provided here :

tdaniele86@gmail.com

orcid -> [0000-0001-5760-7637](https://orcid.org/0000-0001-5760-7637)

curyj@igbmc.fr

orcid -> 0009-0004-7264-9244

mariecharlotte.morin@gmail.com

orcid -> 0009-0001-2993-4283

a.ahier@uq.edu.au

orcid -> 0000-0002-9999-3887

davideisaia5@gmail.com

orcid -> 0009-0004-6337-8956

Specific changes in response to the REVIEWER COMMENTS

Reviewer #1 (Remarks to the Author):

This manuscript offers several notable conceptual and molecular advances in our understanding of the Y-to-PDA transdifferentiation (Td) model and their relationship to Notch signaling dynamics. Overall, the data support a model in which the temporal regulation of Notch activity and that of specific suites of downstream factors enable a sequential series of events such that **(1)** a relatively early pulse of Notch signaling activity in embryogenesis both specifies the Y fate (versus DA9) and triggers the expression of genes required later for Td and **(2)** the timely down-regulation of Notch signaling relaxes the Y fate and allows Td to occur. In particular, the series of experiments expressing Notch NICD from promoters that are (*lin-12p*) or are not (*egl-5p* or *col-34p*) downregulated are compelling. In addition, a new allele of *hlh-16* is identified, the relevant ligands for the Notch receptor are identified, and the mechanism for downregulation of Notch is demonstrated to be transcriptional rather than post-transcriptional or due to changes in ligand availability.

Finally, the authors present copious evidence to counter a previous study suggesting that the timing of Td is affected by Notch signaling levels, providing additional results demonstrating that the *n676n930* allele is not acting in a *gf* manner at 15° as it does in other phenotypic contexts. Rather it acts as a *lf*. Here they argue persuasively, using a battery of markers, that a 2 PDA phenotype is never observed, but rather a 0 Y (and PDA), 2DA9 is observed as would be expected for a loss or reduction

of *lin-12* activity, and indicating with markers that the previously reported precocious PDA was actually DA9.

The data are robust and are presented clearly. While direct demonstration of temporally regulated degradation of *egl-5p* NICD activity (e.g. by degron or by RNAi) and observation of the expected restoration of the 2 PDA phenotype is not presented, sufficient convincing independent approaches are provided to support the model.

Nevertheless, in several places the **text is needlessly confusing and requires revision.**

1. *Title is confusing – particularly “positive selection” and “defined window of opportunity”—and should be changed to reflect the main findings in a more direct way.*

DONE: In response to the referee, we changed the title for: “Essential and dual effects of Notch activity on a natural transdifferentiation event”

2. *Throughout, the authors can be more specific when referring to the previous literature. For example, several places “another cell” or “another player” or “another gene” could just state which cell, player and which gene(s).*

DONE, see lines 58, 164, 538, 584

3. *Heatshock promoter expression is not evenly ubiquitous. This must be stated more carefully.*

DONE: Absolutely - we added this precision as well as the original reference where *hsp-16.2^{Prom}* expression is characterized (see lines 133-137, 268).

4. *The term “Notch signal” should be avoided since it can be misunderstood as a reference to the ligand. “Notch signaling” is better.*

DONE, all relevant instances throughout the main text and legends have been changed (see lines 30, 93, 94, 96, 100, 138, 140, 141, 156, 216, 232, 239, 240, 242, 292, 299, 300, 303, 304, 314, 316, 329, 354, 381, 388, 423, 448, 521, 526, 528, 545, 572, 576, 654, 671)

5. *It is important to mention the incomplete penetrance of the phenotypes in the text as well as in the figures; alert the reader to what they should focus on (e.g., the appearance of any 2 PDA vs all 1 PDA).*

DONE: we have added both in the text when relevant (lines 164, 168, 287), and in the legends (Fig. 1B-F, 3A-B, 4A-J, 5D-G, 7A-G, 8A-G) the notion of penetrance or that a fraction of the worms showed the phenotype

6. *The terms describing gene expression experiments need to be much clearer regarding temporal versus spatial (or both) alterations in gene expression. Throughout the manuscript, the terms “ectopic, exogenous, extrinsic, intrinsic” are often confusing and need to be replaced or better qualified. For example, in cases of heterologous gene expression, it may be best to state the promoter and its presumed temporal and spatial expression pattern.*

DONE: we have suppressed or replaced all 8 instances “exogenous” (see its replacement by “ectopic” lines 32, 128, 138, 231, 292, 299). We have further clearly defined “ectopic” as “experimentally-provided” (line 89). We have used extrinsic and intrinsic to describe WT signals or processes naturally originating, respectively, from outside, or present inside, the cells considered (after verifying in a dictionary). We also have made sure that the first time a specific promoter is used to drive heterologous expression, its expression pattern is described, that the expected cellular expression pattern is spelled out () and we refer to the figure summarising the temporal expression in the Y and rectal cells of the *lin-12*, *egl-5* and *col-34* promoters (Fig 2C and see lines 180 or 286-289).

7. *There is a lot of interesting information about hlh-16 in the section starting line 148; consider a separate section?*

DONE: Great suggestion! We have made it a separate section as recommended (see lines 162-185)

8. *The term dichotomic is very confusing. Even “paradoxical”, which is better, could be recast as “seemingly paradoxical”. However, both are based on the history of the field. Once the model is presented, it is not so paradoxical, but rather all about temporal regulation of lin-12 targets (direct or indirect, maintained or not). The story may be less confusing if it were possible to very clearly present the expectation, the deviation from expectation, the hypothesis, and then the data that support it. That is, based on lin-12p experiments, and the similar onset of expression of egl-5p in all rectal cells including DA9 and Y, you would have expected egl-5p::NICD in Y to have at least 1 PDA phenotype, if not 2 PDA but saw 0 PDA. You interpret 0 PDA as failure of the specified Y to Td into PDA. Given that egl-5p expresses over a longer timeframe than lin-12p, you then hypothesized that the perdurance of NICD could block Td and tested this directly with col-34, etc. Here a “window” concept would be replaced with dynamic temporal expression concept (as is your model), with two sets of targets: (1) early Notch activity and targets specify Y identity (versus DA9) and competence for future Td (2) the necessity to down-regulate the Y-identity signal and allow Td to occur in response to Td-promoting targets triggered by early N. While I appreciate the attempt to mostly provide the data before providing the model, the situation here is sufficiently complex that it would be worth considering the benefit to the reader to suggest the model and then provide evidence. I would also refer the reader to the discussion in the text of the results as the points are further well-developed in the discussion*

DONE: We agree that more clarity was needed to distinguish endogenous Notch action and the outcomes of timed ectopic Notch signalling. We thus have reworded the relevant sections throughout the text according to the referee comments. “Paradoxical” was removed (line 97); We have reduced the use of “dichotomic”, which is not used anymore when the WT process is described (see lines 97, 265, 292-294, 321, 435, 648, 651-653, Figure 9) and was kept to describe the 2 possible outcomes of ectopic Notch signalling, clearly stating that this refers to the actions of experimentally-induced ectopic Notch signalling (eg lines 294, 299, 616), or was changed to “ectopic” (see lines 265, 435). See also response to comment #6.

In addition, we have reworded the section mentioned (p8) to spell out our expectations and logical flow, as recommended by the referee. We also have added a reference to the discussion for in-depth consideration of these points, and reworded the discussion to both remove any redundancy and make our point clearer.

9. *The argument for a “permissive context” (line 224 section) was not clear.*

DONE: We have changed this section to better explicit the notion of permissive context, and have reworked it for a smoother insertion in the rest of the manuscript (see lines 268-283 and 599-618)

10. *I found the introduction of DA9prog and Yprog terms to be confusing. Maybe “future DA9”?*

DONE: We have tried several options to avoid using a heavy peri-sentence to refer to ABplpppaaaa and ABprpppaaaa. In the end, we believe that Y^{prog} and DA9^{prog} best describe that these cells, future rectal Y and DA9 neuron respectively, are still in an undifferentiated progenitor state. Thus, we have remodelled and better introduced our rationale to use these terms describing them clearly as future Y and future DA9 (line 115-117 & line 146-148), and we believe this addresses the referee’s concern.

11. *Related to point above, reword line 553-558, dichotomic, etc. and later in the next paragraph. It is not that Notch activity is blocking Td in the natural context; this only occurs in the experimental situation. Rather, it is promoting Td by activating hlh-16 and sem-4 earlier.*

DONE: see point 8 and lines 612-618 (previously 553-558), we have reworded the relevant sections according to the referee comments: “dichotomic” has been explicated and mostly replaced by

“ectopic”. The difference between endogenous Notch and experimentally-induced Notch is explained explicitly. The next paragraph has also been reworded according to the point and previous ones (lines 639- 669).

12. *The figure presented in SI Fig 3A was extremely helpful to understand the temporal dynamics of the system and the experiments and should be considered as a main figure panel in some form.*

DONE: we have now added this figure as Figure 2C.

13. Additional specific points:

- a. line 51, will transdifferentiate; **DONE**, line 57
- b. line 52, certain cell types; **DONE**, line 58
- c. line 74, replace “make” with “specify”? **DONE**, line 80
- d. line 91, not really a paradoxical action on the Y fate – just paradoxical with respect to the expected phenotypes **DONE**, we have changed this sentence and generally better explained that deregulation of Notch signalling timing can exert a paradoxical action / result in opposite outcomes, lines 96-97
- e. line 94, delete “on” **DONE**, line 100
- f. line 100-101, given points above, reconsider title – ectopic, induce, extra cell **DONE**, lines 106-107
- g. line 113, specify other mutant backgrounds **DONE**, line 120
- h. line 126-127, reconsider use of exogenously, ubiquitously **DONE**, already addressed in points #3 and #6 above, lines 135, 137
- i. line 131, not supernumerary Td, but supernumerary PDA : here and later, supernumerary refers to the number of something, not a process **DONE**, lines 134, 140, 145, 198, 420
- j. line 159, reconsider vague term “assorted” **DONE**, (changed to “concomitant”, line 201)
- k. line 224, reconsider section title **DONE**, we have removed the title altogether and grouped this section with the next one (lines 268-283)
- l. line 243 and section, reconsider “dichotomic” **DONE**, we have changed this title to “Ectopic Notch signalling can both promote and block Y-to-PDA Td”; see also answer to point #8, lines 265-283
- m. line 296, reconsider title to be more clear (Notch activity, “it”) **DONE**, we have changed it to “Td initiation is blocked by late Notch signalling over-imposing a rectal identity”, lines 341-342
- n. line 321, reconsider title **DONE**, we have changed it to “Down-regulation of Notch in the Y cell occurs at the transcriptional level”, line 369
- o. line 384, dichotomic **DONE**, see also answer to point #8 line 435
- p. line 524, “another cell” **DONE**, line 584

Reviewer #2 (Remarks to the Author):

In the provided manuscript, the authors report on how Notch signaling affects the transdifferentiation of rectal epithelial cells in the nematode *C. elegans*.

Natural Transdifferentiation phenomena such as the direct conversion of the rectal epithelial cells Y to the motoneuron PDA in *C. elegans* offer an excellent research system to elucidate the molecular dynamics of cell fate conversion processes. Moreover, this particular natural transdifferentiation instance allows investigations at the single-cell level in vivo with high accuracy. The group of S. Jarriault has been central in dissecting the above-mentioned cell fate conversion process and pioneered a large number of critical findings. In this study, they defined the temporality of Notch signaling, a highly conserved signaling pathway relevant to cell fate specification and many other developmental processes. Interestingly, in previous studies, NOTCH signaling has been implicated in cellular reprogramming as either an inhibitory or promoting pathway depending on the

context of reprogrammed cells and their niche, and species. Yet, this study by Daniele et al. reveals in a very convincing manner that NOTCH has a dichotomic effect on the same cellular reprogramming instance, depending on timing. This outcome is highly relevant for the reprogramming research field and prompts for more careful assessment of cellular signaling pathways that affect reprogramming efficiencies in general.

Moreover, the meticulous work by Daniele et al. in this study also clarifies an observation made by a previous study from Rashid et al. concerning the timing of PDA neuron fate induction in the *lin-12* Notch(n676n930) mutants background that supposedly has a gain-of-function effect at 15°C. Rashid et al., reported that gain of Notch function leads to a precocious Y to PDA conversion. Yet, Daniele et al. demonstrate in a very convincing manner, that the Y cell started expression DA9 cell marker. Examining the DA9 neuron cell marker *mig-1351* they saw that *lin-12* Notch(n676n930) worms raised at the same temperature as in the Rashid et al. study show two DA9 neurons but no Y cell. Hence, they conclude that the progenitor Y cell is mis-specified to a DA9 neuron rather than becoming a PDA neuron precociously.

The study has been performed using an impressively large number of different worm strains with different genetic backgrounds and reporter transgenes, highlighting the effort that has been undertaken to elucidate gene expression and cell fate induction dynamics and temporality with high accuracy.

Overall, this study is of high value for a variety of research fields dealing with development, cellular reprogramming, and Notch signaling. I have only a few comments and questions that should be addressed before publication.

Comments and questions to the authors:

1. *Line 113 – 114: Explain or mention briefly “...other mutant backgrounds do not have the capacity to transdifferentiate into a PDA neuron.”*

DONE, we have now explicitized which mutant backgrounds exhibit an additional Y cell that is not competent to transdifferentiate, as shown in Jarriault et al 2008 (see line 120)

2. *Figure 1: 3-Fold embryo in panel G has very limited capacity to provide any information – near useless as the movement of the animal causes blurring and mixing of signals during the exposure time. The use of a bright GFP-based injection marker combined with the actual GFP-reporter is very unfortunate. A workaround instead of making a new strain could be to fix the animals briefly and / or stain them with anti-GFP Abs.*

DONE, Thank you for pointing this out, we have replaced this picture by a better one (now Fig. 2A)

3. *Strain list: The long list of worm strains used is impressive and at the same time makes it difficult to easily find information about strains and transgenes. E.g. what is the exact nature of the *hlh-16::gfp* reporter?*

DONE, Thank you for the praise! We have added additional info both in the M&M and in the Supplementary Information. In addition to the plasmid descriptions in the M&M, we have added 2 novel Supplementary tables, for Ex and Is arrays. For each transgene generated in the lab, the tables contain columns for the injection marker, as well as the relevant transgene content and the name of the plasmid used, and the genetic background in which it was injected. We have also clearly labelled the transgenes that were generated in the lab.

4. *Notch signaling:*

A. *Is it sufficient to have Notch only during embryo development (depleted starting L1) to allow Y to PDA conversion?*

DONE: We thank the referee for this insightful question. We had addressed it in a previous publication (Jarriault et al. 2008) using a *lin-12(lf)* temperature sensitive allele, as a *lin-12(lf); hs::NICD* strain turned out to be too sick to make. We showed that when the *lin-12(n676n930ts)*

strain was shifted to the restricted temperature (25°C) after the 3-fold stage (ie no Notch signalling anymore), a wild type Y-to-PDA Td was observed (note that Notch receptor half-life is very short). These results strongly suggest that Notch presence and activity is not necessary for Y-to-PDA from late embryogenesis on. Our recent data presented here support this: a short pulse of expression of NICD specifically in the embryo is sufficient to induce an additional Td event. In addition, our data expressing the WT LIN-12 full length receptor under *egl-5* promoter (ie, continuously in Y^{prog} and the rectal Y cell) show that LIN-12 continued presence in Y after its normal expression timing is detrimental : when it is re-expressed in the rectal Y cell up to Td initiation timing, as its ligands are still present around and active, its presence results in a block of Y Td.

B. Is the NICD always transported to the nucleus – this may be important to know whenever NICD (over)expression does not show an effect.

DONE, this is an important control that we have now added as Supplementary Fig. 3 (pictures and quantifications): indeed, when NICD is ectopically expressed in Y or in the rectal cells, we have found it to always be nuclear, suggesting that its inability to promote transdifferentiation of the other rectal cells into a PDA neuron is not linked to its ability to enter/remain in the nucleus.

C. The question before and other matters could be addressed if something like a ‘Notch sensor’ is possible. Meaning, a reporter construct that bears several copies of binding sites for the Notch effector protein. This reporter could be used to monitor whether the examined cells and time points are indeed ‘receiving’ effective Notch signaling (also useful for potential non-cell-autonomous effects).

DONE: Great suggestion: indeed, we had tried to develop multicopy Notch activity reporters using multimerised *lag-1* binding sites in different orientations and numbers. However, these never worked, such that translocation to the NY remains our best tool so far. As explained above (see point #4B), we have quantified NICD localisation in our OE strains (Supplementary Fig. 3) and found it to be always nuclear.

5. Figure 5: Panels A,E axis labels are hard to read – too small!

DONE, Thank you for pointing this out. We have homogenised the size of the font on all figures following Nature Communications guideline.

6. Line 375 -377 Statement about Notch conserved domain appears misplaced in this study.

DONE, We have replaced the data on these two conserved regions in the promoter of *lin-12* in the section dedicated to *lin-12* transcriptional regulation, and we altered the sentence to make it clearer that we are not referring to domains in LIN-12 protein (see lines 426-427).

7. Line 399 The sentence reads difficult – ‘made’ at the end of the sentence, correct?

DONE, we have modified the sentence accordingly (lines 450).

8. Some parts of the discussion, like at the beginning, are too long or redundant with other sections. Overall, it would help the manuscript to be a bit more concise.

DONE, we have rewritten parts of the discussion to remove any redundancy and have shortened it.

REVIEWERS' COMMENTS

Reviewer #1 (Remarks to the Author):

The authors have addressed well all previously listed concerns.

Reviewer #2 (Remarks to the Author):

I do not have any further comments or suggestions for corrections.

Congratulations on this exquisite work. I fully support the publication.